# Constraining a complex biogeochemical model for multi-site greenhouse gas emission simulations by model-data fusion

Tobias Houska[1], David Kraus[2], Ralf Kiese[2], and Lutz Breuer[1,3]

[1] Institute for Landscape Ecology and Resources Management (ILR), Research Centre for BioSystems, Land Use and Nutrition (iFZ), Justus Liebig University Giessen, Giessen, 35392, Germany

[2] Institute of Meteorology and Climate Research - Atmospheric Environmental Research (IMK-IFU), Garmisch-Partenkirchen, 82467, Germany

[3] Centre for International Development and Environmental Research (ZEU), Justus Liebig University Giessen, Giessen, 35392, Germany

*Correspondence to*: Tobias Houska (tobias.houska@umwelt.uni-giessen.de)

**Abstract:** This study presents the results of a combined measurement and modelling strategy to analyse $N_2O$ and $CO_2$ emissions from adjacent arable land, forest and grassland sites in Germany. The measured emissions reveal seasonal patterns and management effects, including fertilizer application, tillage, harvest and grazing. The measured annual $N_2O$ fluxes are 4.5, 0.4 and 0.1 kg N ha$^{-1}$ a$^{-1}$, and the $CO_2$ fluxes are 20.0, 12.2 and 3.0 t C ha$^{-1}$ a$^{-1}$ for the arable land, grassland and forest sites, respectively. An innovative model-data fusion concept based on a multi-criteria evaluation (soil moisture at different depths, yield, $CO_2$ and $N_2O$ emissions) is used to rigorously test the LandscapeDNDC biogeochemical model. The model is run in a Latin Hypercube based uncertainty analysis framework to constrain model parameter uncertainty and derive behavioural model runs. The results indicate that the model is generally capable of predicting trace gas emissions, as evaluated with RMSE as the objective function. The model shows a reasonable performance in simulating the ecosystem C and N balances. The model-data fusion concept helps to detect remaining model errors, such as missing (e.g., freeze-thaw cycling) or incomplete model processes (e.g., respiration rates after harvest). This concept further elucidates the identification of missing model input sources (e.g., the uptake of N through shallow groundwater on grassland during the vegetation period) and uncertainty in the measured validation data (e.g., forest $N_2O$ emissions in winter months). Guidance is provided to improve the model structure and field measurements to further advance landscape-scale model predictions.

## 1 Introduction

Carbon dioxide ($CO_2$) and nitrous oxide ($N_2O$) are two prominent greenhouse gases (GHG) contributing to global warming, the latter having a global warming potential (GWP) 300 times higher than that of $CO_2$ considering a 100-year time horizon (Myhre et al., 2013). Terrestrial ecosystems play an important role in the global atmospheric budgets of both GHGs (Cole et al., 1997). The global $CO_2$ emissions from soils are five times higher than anthropogenic (mainly fossil fuel) $CO_2$ emissions (Raich and Schlesinger, 1992; updated with recent fossil fuel data by Boden, et al., 2010), while agricultural land use released over 60% of the global anthropogenic $N_2O$ emissions in 2005 (IPCC, 2007). In addition to the radiative forcing of both GHGs, $N_2O$ is currently the main driver of stratospheric ozone depletion (Ravishankara et al., 2009), causing increased ultraviolet

radiation, which could result in skin cancer and other health problems (Graedel and Crutzen, 1989). While $CO_2$ is exchanged with the soil (heterotrophic respiration) and vegetation (photosynthesis and autotrophic respiration), $N_2O$ fluxes refer mainly to the nitrification and denitrification processes occurring only in the soil (Butterbach-Bahl et al., 2013).

Emissions of both GHGs are highly variable in space and time and depend on a multitude of different interacting environmental factors, e.g., land use/management, nitrogen/carbon inputs, meteorological conditions and physical and chemical soil properties (Davidson, 1992; Smith et al., 2003). They are largely regulated by plant physiological (Rochette et al., 1999) and microbial processes (Burton et al., 2008). Field measurements of GHG emissions and environmental drivers have paved the way for a basic understanding of observed emissions patterns. Nevertheless, the large number and complexity of the processes involved in the production and consumption of $CO_2$ and $N_2O$ are still challenges in the reliable quantification of related GHG emissions (Butterbach-Bahl et al., 2013). Various biogeochemical models have been developed in recent years. These models are used for temporal as well as spatial up-scaling of GHG emissions, hypothesis testing of our understanding of processes, and, for scenario analyses and the evaluation of efficient mitigation options (Kim et al., 2015; Molina-Herrera et al., 2016). These include, e.g., BASFOR (Oijen et al., 2005), CERES-EGC (Gabrielle et al., 2006), COUP (Jansson, 2012), DAYCENT (Parton et al., 1998) and DNDC and its descendant LandscapeDNDC (Haas et al., 2013). However, models are still simplifications of the real world and are prone to multiple sources of uncertainty, i.e., defective model structure and/or parameterization and the current model state (Vrugt, 2016). During model application, poor-quality model forcing data results in further uncertainties about the predicted model outcome (Kavetski et al., 2006). However, there is still no method available to properly address these sources of uncertainty at the same time (Vrugt, 2016). One promising way to reduce the magnitude of uncertainties in model output is to use model-data fusion techniques, i.e., matching model prediction with multiple observations by varying model parameters or states using statistical uncertainty estimation (Keenan et al., 2011). There are several statistical uncertainty estimation methods available, e.g., formal Bayesian approaches such as DREAM (Vrugt, 2016) and informal Bayesian approaches such as GLUE (Beven and Binley, 1992). However, these approaches are mostly used to fit models to single types of observations (Giltrap et al., 2010). Innovative multiple observation data evaluations with model-data fusion are becoming common in ecosystem carbon modelling (Wang et al., 2009) and are more and more important in the nitrogen modelling community (Wang and Chen, 2012). The knowledge gained can and should be used to guide further model improvements (Vrugt, 2016).

This work focuses on establishing model-data fusion in the biogeochemical community – i.e., showing the capability of this technique to improve process understanding through the application of process-based models. We present weekly measurements of $CO_2$ and $N_2O$ emissions from a developed landscape with different land uses, i.e., arable land, grassland and forest ecosystems, covering a two-year period of observations. In addition to field measurements, we set up the biogeochemical LandscapeDNDC model for each of the three land uses. During model-data fusion with GLUE, we rigorously accept only model runs that return concurrent, acceptable outputs for $N_2O$, $CO_2$, and soil moisture at different depths and yields. Posterior model runs are not only evaluated as to whether they fulfil appropriate objective functions but also regarding realistic simulations of GHG emissions for separate seasons, annual sums as well as before and after land management. The model is

finally used to estimate the magnitude and uncertainty of C and N fluxes, such as $N_2$ emissions or autotrophic and heterotrophic $CO_2$ emissions, which are not yet experimentally quantifiable *in situ*. The remaining model and data errors are traced back to their potential sources to improve ongoing measurements and future model applications.

## 2 Materials and methods

### 2.1 Study area

The study area is located in the catchment of a low mountainous creek (Vollnkirchener Bach) in the municipality of Hüttenberg, Hesse, Germany (50°29′56″ N, 8°33′2″ E). One kilometre north of the village of Vollnkirchen, next to the creek, we established eight transects (oriented mostly vertically to slope) along a valley cross-section covering different types of land uses (Fig. 1) for GHG emission measurements. See Table 1 for detailed information on soils characteristics. Three transects (A1-A3) are located on arable land to the west of the creek and were cultivated with the same field management and crop rotations (Table 1). Three transects are located in a light beech (*Fagus sylvatica*) forest (W1-W3) with young and old trees on a steep hillside (slope: 10%) east of the creek. A shallow 0.05 m litter layer characterizes the forest soils. Furthermore, there are two transects (G1, G2) located in the riparian zone at a 4 m distance to each side to the Vollnkirchener Bach. One of the two transects is managed and grazed grassland (G1), mainly covered with brown knapweed (*Centaurea jacea*), meadow foxtail (*Alopecurus pratensis*), red clover (*Trifolium pratense*) and ribwort plantain (*Plantago lanceolata*). The second transect (G2) represents a wetland and is mainly covered by meadowsweet (*Filipendula ulmaria*), common nettle (*Urtica dioica*), hoary ragwort (*Senecio erucifolius*) and field bindweed (*Convolvulus arvensis*). The groundwater table is close to the surface on both grassland sites. The mean annual wet depositions of nitrate and ammonium were measured from 2013–2015 with 1.66 kg N ha$^{-1}$ and 3.45 kg N ha$^{-1}$, respectively. In the catchment, the mean annual precipitation is 588 mm, and the mean annual temperature is 10.5 °C for the hydrological year 1$^{st}$ Nov. 2013 - 31$^{st}$ Oct. 2014 (Seifert et al., 2016). The soil moisture is measured at A3 [0.2, 0.4 and 0.6 m], at G2 [0.1 and 0.25 m] and at W1 [0.15 and 0.25 m] and has been recorded at an hourly resolution since 2013. The weekly trace gas measurements began in November 2013 and range so far until December 2015. GHG exchange fluxes were measured manually with non-steady state opaque chambers, each covering a basal area of 0.12 m$^2$. Chambers were placed on frames (both polypropylene), which were inserted approx. 8 cm into the soil in order to facilitate gas-tight sampling as well as to avoid soil structural damage and lateral trace gas leakage. Each chamber is equipped with an extraction septum, a counterbalance valve (in-box pressure balance) and a small fan/ventilator for homogenous mixing of the headspace air. During a 40-minute closure period, five air samples are taken from the chamber headspace at regular time intervals t0-t4 of ten minutes (0, 10, 20, 30 and 40 min.). Samples are analysed by gas chromatography (GC 8610C, SRI Instruments, Torrance, US) with an ECD for $N_2O$ and a methanizer and FID for $CO_2$. Sampling was performed on a weekly basis, with five replicated chambers per transect sampled by the gas sample pooling technique (Arias-Navarro et al., 2013). According to this approach, at any time interval (t0-t4), 10 ml headspace samples are collected subsequently from any of the five replicated chambers and are pooled into one gas-tight glass vial (SRI Instruments). The trace gas fluxes are calculated from the rate of change in the headspace

gas concentration over time by linear regression and were corrected for the chamber temperature, atmospheric pressure and chamber volume according to Barton et al. (2008). All measurements with a regression quality of $r^2 < 0.7$ for $CO_2$ (using at least four individual samples) were rejected.

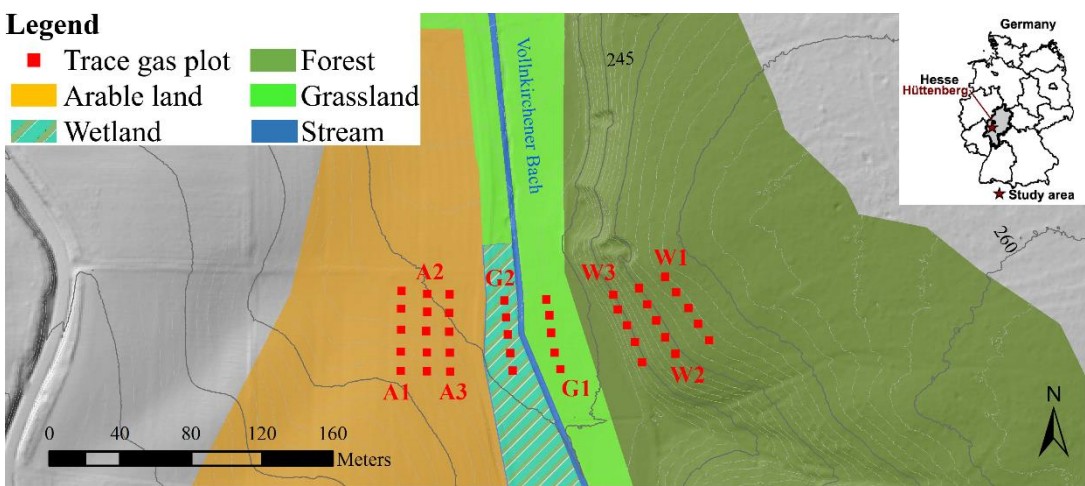

**Figure 1: Map of the study area. Red squares represent GHG chamber positions at the different transects. Dark grey contour lines represent 5 m differences in elevation. Light grey areas are outside of the catchment area.**

Soil emissions of $CO_2$ and $N_2O$ can be subject to significant diurnal patterns, with peak values observed in the early afternoon (Savage et al., 2014), impeding the up-scaling of hourly measured emissions (usually obtained at midday) to daily values. We performed multiple linear regression (ordinary least squares regression including air temperature, relative humidity and water filled pores space) to account for the difference between, e.g., daytime (Wohlfahrt et al., 2005a) and night-time respiration (Wohlfahrt et al., 2005b). In our dataset, only $CO_2$ emissions showed significant correlations with the mentioned environmental drivers on arable land ($r^2 = 0.53$), grassland ($r^2 = 0.59$) and forest ($r^2 = 0.51$). Following Subke et al. (2003), we derived an hourly integration formula in order to obtain daily representative mean values of $CO_2$ emissions from our field measurements conducted mostly between 9 am and 5 pm. $N_2O$ emissions are up-scaled to daily mean values with the common approach, i.e., by multiplying hourly emissions by 24. Annual $CO_2$ and $N_2O$ emissions are calculated by linear interpolation between the measurements. All the underlying data in sections 2.1 and 2.2 are available upon request from a database (http://fb09-pasig.umwelt.uni-giessen.de:8081/).

### 2.3 Modelling approach

### 2.3.1 Model set up

We tested the biogeochemical model framework LandscapeDNDC (Haas et al., 2013) with the observed data from our study area. Individual models were set up for arable land, grassland and forest ecosystems. The models describe different processes in ecosystem compartments, i.e., mathematical descriptions of microclimate, water cycle, plant physiology and soil

biogeochemical processes. We applied the biogeochemical model MeTr$^x$ (Kraus et al., 2015) and the water cycle model watercycleDNDC (Kiese et al., 2011) for all land uses. The biogeochemical model MeTr$^x$ simulates the turnover of soil organic matter and plant debris depending on their chemical structures (e.g., lignin and cellulose content, C/N ratio), soil properties (e.g., pH value) and meteorological drivers. Following the 'anaerobic balloon' concept of Li et al. (2000), major metabolites

(e.g., $NO_3$) are distinguished between aerobic and anaerobic counterparts in order to simulate the share of nitrification and denitrification and the related production of GHG emissions. Simulated model outputs are, among others, emissions of $CO_2$ and $N_2O$. The watercycleDNDC model simulates soil water dynamics, i.e., potential evapotranspiration based on Thornthwaite and Mather (1957), transpiration depending on gross primary productivity, the water use efficiency of the modelled plant types and soil water flow based on a cascading bucket model approach (Kiese et al., 2011). The latter determines the advective

transport of nutrients into deeper soil layers.

All models refer to a one-dimensional soil column, i.e., assuming homogeneous conditions in lateral directions, and were run with a daily time step resolution. Tab. 1 provides an overview of the major model driving data, i.e., meteorological data and land use-specific soil and vegetation characteristics. To simulate plant growth on the three different land use types, we selected the individual physiology modules arableDNDC, grasslandDNDC (Kim et al., 2015; Molina-Herrera et al., 2016) and PSIM

(Grote et al., 2009).

Arable soils are stagnic luvisols with a thick loess layer, modelled down to 2.0 m with 80 layers, while the actual soil depth is unknown. Gleysols in the meadow grassland site were modelled down to 0.5 m (set up with 40 layers), corresponding to the mean annual groundwater table depth. The thin and stony soil at the forest site is a cambisol and modelled down to bedrock (0.55 m, set up with 45 layers) with a litter height of 0.05 m. The bulk density increases with depth for every land use, while

soil organic carbon and nitrogen decrease with depth. We run simulations for all land uses at a daily time resolution for 6 years, starting on 1$^{st}$ January 2010, using the data from Table 1 as initialization and using a model spin-up time of two years.

**Table 1: Input settings of the LandscapeDNDC model for the three different land uses in the Vollnkirchener study region, based on measurements and farmers management documentation. In case spans are given, they reflect observed ranges for measurements used throughout the set up of the soil profile, given from the top layer setting to the bottom layer. The soil depth was estimated for model set up. F = fertilizer application, M = manure application.**

| Input | Arable (A1-3) | | | | Grassland (G1) | | Forest (W1-3) | Unit |
|---|---|---|---|---|---|---|---|---|
| Vegetation type | Sep 10 - Jul 11 | Winter Barley | | | Perennial grass | | Light beech forest | - |
| | Aug 11 - Aug 12 | Rape | | | | | | |
| | Oct 12 - Aug 13 | Winter Wheat | | | | | | |
| | Oct 13 - Aug 14 | Triticale | | | | | | |
| | Sep 14 - Aug 15 | Triticale | | | | | | |
| | Oct 15 - Jul 16 | Rape | | | | | | |
| Management | 2 Mar12 | 166.5 | kg N ha$^{-1}$ | (F) | 01 Feb 13 | Grazing | | - |
| | 2 Apr 12 | 49.9 | kg N ha$^{-1}$ | (F) | 01 May13 | Harvest | | |
| | 8 Nov 12 | 56.2 | kg N ha$^{-1}$ | (F) | 01 Sep 13 | Grazing | | |
| | 11 Mar 13 | 54.0 | kg N ha$^{-1}$ | (F) | 02 Mar 14 | Grazing | | |
| | 23 Apr 13 | 53.8 | kg N ha$^{-1}$ | (F) | 01 May 14 | Harvest | | |
| | 3 May 13 | 29.3 | kg N ha$^{-1}$ | (M) | 01 Sep 14 | Grazing | | |
| | 3 May 13 | 538.0 | kg C ha$^{-1}$ | (M) | 20 Jan 15 | Grazing | | |
| | 12 Nov 13 | 29.0 | kg N ha$^{-1}$ | (M) | 29 Jun 15 | Harvest | | |
| | 12 Nov 13 | 533.0 | kg C ha$^{-1}$ | (M) | 26 Sep 15 | Grazing | | |
| | 11 Mar 14 | 54.0 | kg N ha$^{-1}$ | (F) | | | | |
| | 1 Apr 14 | 53.8 | kg N ha$^{-1}$ | (F) | | | | |
| | 8 May 14 | 40.5 | kg N ha$^{-1}$ | (F) | | | | |
| | 22 Sep 14 | 149.0 | kg C ha$^{-1}$ | (M) | | | | |
| | 22 Sep 14 | 8.1 | kg N ha$^{-1}$ | (M) | | | | |
| | 8 Nov 14 | 1032.0 | kg C ha$^{-1}$ | (M) | | | | |
| | 8 Nov 14 | 56.2 | kg N ha$^{-1}$ | (M) | | | | |
| | 11 Mar 15 | 1564.0 | kg C ha$^{-1}$ | (M) | | | | |
| | 11 Mar 15 | 85.1 | kg N ha$^{-1}$ | (M) | | | | |
| | 10 Apr 15 | 59.4 | kg N ha$^{-1}$ | (F) | | | | |
| | 30 Aug 15 | 59.4 | kg N ha$^{-1}$ | (F) | | | | |
| | 12 Nov 15 | 29.0 | kg N ha$^{-1}$ | (M) | | | | |
| | 12 Nov 15 | 532.0 | kg C ha$^{-1}$ | (M) | | | | |
| Soil texture | Sandy clay loam | | | | Sandy clay loam | | Sandy clay loam | - |
| Soil type | Stagnic Luvisol | | | | Gleysol | | Cambisol | |
| Bulk density | 1.55–1.60 | | | | 1.20–1.44 | | 1.36–1.49 | g cm$^{-3}$ |
| Organic carbon | 1.57–0.91 | | | | 2.55–0.71 | | 3.61–1.73 | % |
| Total soil nitrogen | 0.16–0.09 | | | | 0.29–0.08 | | 0.21–0.11 | % |
| Clay content | 23–26 | | | | 24–25 | | 24–26 | % |
| pH | 6.45 | | | | 4.42 | | 3.5–5.5 | - |
| Soil depth | 2.00 | | | | 0.50 | | 0.55 | m |

### 2.3.2 Model-data fusion

For the multi-objective Bayesian model calibration, we used a two-tiered Generalized Likelihood Uncertainty Estimation (GLUE) approach (Beven and Binley, 1992). The model was iterated in both tiers 100,000 times by changing the parameter sets using Latin hypercube sampling with the Python software SPOTPY (Houska et al., 2015). The parameters for the physiology and the water-cycle modules were treated as land use-specific, while the parameters of the biogeochemical model were calibrated using the data from all land uses (Table A1). We presuppose no prior knowledge besides the given parameter ranges, i.e., we assume a uniform (non-informative) prior probability distribution for all parameters. We statistically judged the performance of every parameter set to reproduce measurements with a root mean squared error (RMSE). Similar to Bloom and Williams (2015), we do not explicitly consider measurement uncertainty during the model data fusion. As shown in Houska et al. (2017), one-tier GLUE based multi-objective model calibration can result in very low acceptance rates, down to 0.01%. We therefore considered a two-tier GLUE approach in order to increase the identifiability and accuracy of the accepted model runs:

Tier I: In the first step, we constrained the parameter space of the hydrology and plant physiology modules of LandscapeDNDC by investigating the respective parameters of both models (Table A1). We accepted only model runs that were within the best 5% of all simulated RMSEs in terms of the respective variable (WFPS at different depths [arable land at 0.2, 0.4 and 0.6 m, grassland at 0.1 and 0.25 m and forest at 0.15 and 0.25 m], as well as yield on arable land). Parameter sets were accepted if they belonged to the 5% best model runs for each land use. That is, we took the best 5% of the RMSEs for each respective output variable and took only the intersecting parameter set, which are all from the selected variables for one land use. The results of tier I are summarized in supplementary Fig. A1-A4 and are not further discussed in this study, as they belong to the initialization of the model.

Tier II: To achieve realistic GHG simulations from the MeTr$^x$ biogeochemical module of LandscapeDNDC, we took the posterior parameter boundaries of tier I and ran GLUE with all parameters of Table A1 again. This time, we considered the best 5% of all RMSEs in terms of the respective $N_2O$ and $CO_2$ emissions for each land use (A1-3, G1 and W1-3). Again, only the 5% best intersecting parameter sets were accepted per land use. These results are shown in the following chapters. There was no major effect of the biogeochemical model parameters on the WFPS simulation.

Posterior model runs of tier II were further investigated in three different ways:

(1) Seasonal comparisons of measured and modelled emissions for spring (21$^{st}$ March - 20$^{th}$ June), summer (21$^{st}$ June - 20$^{th}$ September), autumn (21$^{st}$ September - 20$^{th}$ December), and winter (21$^{st}$ December - 20$^{th}$ March).

(2) Management comparison of measured and modelled emissions, i.e., investigation of model performance within two weeks before and two weeks after management events to check model performance in generating hot moments, e.g., after fertilizer application.

(3) Model performance in simulating magnitude and uncertainty of C and N fluxes not measured *in situ*, such as $N_2$ or autotrophic and heterotrophic components of $CO_2$ emissions.

## 3 Results and discussion

### 3.1 Measured $N_2O$ fluxes

To determine the representativeness of each transect for a given land use, the respective differences in measured $N_2O$ emissions were compared (Table 2). The temporal dynamics of $N_2O$ emissions are presented (Fig. 2), distinguishing between different seasons (Fig. 3) and before/after management events (Fig. 4).

Arable land $N_2O$ fluxes: Emissions on arable land vary between 0 and 0.3 kg $N_2O$-N ha$^{-1}$ day$^{-1}$. There were no significant differences over time between the three weekly measured transects on arable land (Table 2). The highest emissions occur

mostly after management events. Mineral fertilizer application in particular stimulates $N_2O$ emissions, causing hot moments from, for example, March to May 2014. The input of N through manure application has a minor influence on the magnitude of $N_2O$ emissions. The mean annual measured $N_2O$ emissions from arable land are comparably high with 4.5 kg $N_2O$-N ha$^{-1}$ a$^{-1}$ (Jungkunst et al., 2006), equalling a GWP of 575 kg $CO_2$-C equiv. ha$^{-1}$ a$^{-1}$. With a yearly fertilizer application of 248.2 kg N a$^{-1}$ a mean annual emission factor (EF) of 1.4% (varying between 1.2% for A2 and 1.8% for A3) can be calculated, where

1 kg N ha$^{-1}$ a$^{-1}$ is attributed to the background emissions of unfertilized soil (IPCC, 1997). This EF is inside the IPCC-assumed range of 1.25 $\pm$1% and close to the average EF (1.56%) of several (n=56) agricultural sites in Germany (Jungkunst et al., 2006). A robust finding throughout the literature is that reduced nitrogen input would lead to lower emissions and therefore more climate-friendly agriculture (Bouwman et al., 2002).

Grassland $N_2O$ fluxes: $N_2O$ emissions significantly vary between the grazed site G1 and the wetland site G2, which can be

attributed to differences in management, hydrological, soil and vegetation characteristics. Most likely, the nitrate supply through groundwater and uptake by the rooting system of the plants is important (Liebermann et al., 2017). Even though the groundwater table (0.2 - 0.4 m belowground) is rather shallow in the winter/spring, the uptake rates in summer/autumn (groundwater table 0.3 - 1.0 m belowground) are supposedly larger due to the vegetation period. Here, capillary rise may play a relevant role (Orlowski et al., 2016). G1 is characterized by a mix of *Centaurea jacea*, *Alopecurus pratensis*, *Plantago*

*lanceolata* and *Trifolium pratense,* is grazed by sheep twice a year and is cut once a year. Emissions from the grazed grassland vary between -0.0019 and 0.014 kg N ha$^{-1}$ day$^{-1}$. High emissions were measured after grazing, e.g., in March 2014 when sheep dung was stimulating $N_2O$ emissions. Negative values depict $N_2O$ uptake and are frequently found under prevailing wet conditions in spring, a finding that was also reported by Glatzel and Stahr (2001). The grassland annual $N_2O$ emissions are much lower than those observed for the arable system (A1-3). However, with 0.29 kg $N_2O$-N ha$^{-1}$ a$^{-1}$ are they in accordance

with a study site 12 km northeast of our site, where annual emissions range from 0.18 to 0.79 kg $N_2O$-N ha$^{-1}$ a$^{-1}$ on an unfertilized grassland with shallow groundwater table (Kammann et al., 1998). Their study also reports a similar seasonal pattern to our measurements, with emissions close to zero in the dry and colder autumn months. The measured annual emissions

Table 2: Mean measured annual fluxes (Nov 2013 - Dec 2015) on the different land use transects of the Vollnkirchener Bach study area. Differences between the investigated transects and land uses for measured and modelled $N_2O$ emissions in kg $N-N_2O$ ha$^{-1}$ a$^{-1}$. * = significant difference ($p < 0.05$, Kruskal-Wallis test). Arable (A1-3), Grassland (G1), Wetland (G2), Forest (W1-3), RMSE in kg $N-N_2O$ ha$^{-1}$ day$^{-1}$.

|    | A1 | A2 | A3 | G1 | G2 | W1 | W2 | Measured | Mean measured | Mean simulated | Posterior RMSE |
|----|----|----|----|----|----|----|----|----------|---------------|----------------|----------------|
| A1 |    |    |    |    |    |    |    | 4.08 |      |      | 0.0326 - 0.0353 |
| A2 |    |    |    |    |    |    |    | 3.87 | 4.49 | 7.33 | 0.0238 - 0.0278 |
| A3 |    |    |    |    |    |    |    | 5.53 |      |      | 0.0285 - 0.0329 |
| G1 | *  | *  | *  |    |    |    |    | 0.29 | 0.29 | 0.69 | 0.0029 - 0.0038 |
| G2 | *  | *  | *  | *  |    |    |    | 0.52 | 0.52 | -    | not simulated |
| W1 | *  | *  | *  |    | *  |    |    | 0.09 |      |      | 0.0022 - 0.0025 |
| W2 | *  | *  | *  | *  | *  |    |    | 0.03 | 0.08 | 0.33 | 0.0014 - 0.0021 |
| W3 | *  | *  | *  | *  | *  |    | *  | 0.13 |      |      | 0.0018 - 0.0021 |

are below the assumed background level of $N_2O$-N emissions of 1 kg $N_2O$-N ha$^{-1}$ a$^{-1}$ from agricultural soils (IPCC, 1997). The annual $N_2O$ emissions are equal to a GWP of 37 kg $CO_2$-C equiv. ha$^{-1}$ a$^{-1}$. The EF through grazing is 3.8%, which is in accordance with typical emissions factors from extensive grazed grasslands, ranging globally from 0.2 - 9.9% (Oenema et al., 1997).

Wetland $N_2O$ fluxes: The non-managed transect G2 is dominated by species such as *Urtica dioica*, *Filipendula ulmaria* and *Senecio erucifolius.* Typically, a deeper rooting system is found compared to that in the grazed grassland transect G1, and accordingly, additional nitrate uptake from the groundwater is more prevalent. The mean measured emissions are higher on the non-managed G2 than on the grazed G1 throughout the year, especially during summer and autumn (Fig. 3). The annual emissions are accordingly nearly two times higher at 0.52 kg $N_2O$-N ha$^{-1}$ a$^{-1}$, which is equal to a GWP of 66 kg $CO_2$-C equiv. ha$^{-1}$ a$^{-1}$.

Forest $N_2O$ fluxes: Significant differences were found for the forest transects W2 and W3, which can be explained by natural variations along the steep hillslope: On the hillside (W2) the soil is potentially washed out through lateral transport, leading to decreased nutrient availability, compared to the drier top (W1, +200% $N_2O$ emissions) and the wetter hillfoot (W3, +330% $N_2O$ emissions). The $N_2O$ emissions from the forest transects are mostly low, ranging between -0.003 and 0.004 kg N ha$^{-1}$ day$^{-1}$. Higher emissions were measured only for several weeks in January 2014, with the highest values observed at W1. We attribute this to freeze-thaw effects, typically found when year-around measurements are considered (Papen and Butterbach‑Bahl, 1999). Negative fluxes were measured, for example in March and May 2014. The underlying process of $N_2O$ uptake has been reported before (e.g., Flechard et al., 2005; Neftel et al., 2007) and is assumed to be a microbial process, in which denitrifiers use $N_2O$ as an electron acceptor for respiration under wet/anaerobic conditions (Bremner, 1997). Negative emissions occur during times with high WFPS (Fig. A3), which is in accordance with Bremner (1997). However, our measured negative emissions are low compared to the variance between transects (W1-3), i.e., they could also originate from measurement errors. Our annual measured emissions in forests are 0.08 kg $N_2O$-N ha$^{-1}$ a$^{-1}$ (GWP of 10 kg $CO_2$-C equiv. ha$^{-1}$ a$^{-1}$ $CO_2$ emissions), which is much lower than that at adjacent grassland and arable sites. Moreover, this value is almost two orders of magnitude lower than the $N_2O$ emissions (5.1 kg $N_2O$-N ha$^{-1}$ a$^{-1}$) measured from a beech

forest in Högelwald, Germany (Papen and Butterbach-Bahl, 1999). A likely reason is the substantially higher annual deposition rate of 25 kg N ha$^{-1}$ a$^{-1}$, an N input five times higher than that in our system. However, our measurements of N deposition only include wet deposition. Additional dry depositions are often assumed to add another 30-60% to total atmospheric N deposition (Flechard et al., 2011).

## 3.2 Measured $CO_2$ fluxes

Emissions measured using our closed chamber on arable land and grassland include those from soil and vegetation, as entire plants are covered by the chamber. Therefore, we interpret these emissions as total ecosystem respiration (TER). In contrast, chambers in the forest were placed on the forest floor without any vegetation inside; thus, these measurements include soil (heterotrophic) and root (autotrophic) respiration, i.e., below ground respiration only. To determine the representativeness of

10 each transect for a given land use, the respective differences in measured $CO_2$ emissions were compared to each other (Table 3). The measured $CO_2$ emissions are given over time (Fig. 5), separated into different seasons (Fig. 6) and before/after management-events occur (Fig. 7).

Arable TER: Measured values from our arable transects range between 0 to 175.2, 199.6 and 143.1 kg C-$CO_2$ ha$^{-1}$ day$^{-1}$ for A1, A2 and A3 respectively and are not significantly different between the transects (compare Table 3). Emissions occur

mainly during the growing season, starting in March and ending in November. For a comparable study site in southern Finland, reported daily TER values under barley were between 23.6 to 235.6 kg C-$CO_2$ ha$^{-1}$ day$^{-1}$ during May and September (Lohila et al., 2003), which is in the same range as our observations. The annual sum of our TER emissions is $19.96 \pm 2.36$ t C-$CO_2$ ha$^{-1}$ a$^{-1}$. This is slightly lower than yearly TER measured on a winter wheat study site in Belgium with 23.18 t C-$CO_2$ ha$^{-1}$ a$^{-1}$ (Suleau et al., 2011). Demyan et al. (2016) reported lower values, with an average total of

11.43 t C-$CO_2$ ha$^{-1}$ a$^{-1}$, derived from observations spanning six growing seasons in southwestern Germany. However, all studies are possibly prone to overestimations of the emissions from September to November, as daily emissions are generated with a multiple linear regression model, and in our case, are based on our hourly measurements of air temperature and soil moisture. Such methods do not fully account for management effects, such as harvests (Subke et al., 2003).

Grassland TER: Emissions from grassland vary from 5.0 to 68.3 t C-$CO_2$ ha$^{-1}$ a$^{-1}$, with no significant difference between the

25 two transects G1 and G2. Emissions are close to zero in the winter months (December to February) and highest during the growing season. A distinct negative correlation between the measured TER with WFPS was found during wet conditions from end of June to July in 2014. In this time, emissions decrease to 41.0 kg C-$CO_2$ ha$^{-1}$ day$^{-1}$. The total yearly emissions are 11.79 t C-$CO_2$ ha$^{-1}$ a$^{-1}$, which agrees well with the mean yearly emissions reported for 19 different grassland sites across Europe, with mean annual emissions of 12.83 t C-$CO_2$ ha$^{-1}$ a$^{-1}$ (Gilmanov et al., 2007). However, due to the many different

grassland sites considered in their study, Gilmanov et al. report a much wider range of observed annual TER values, from 4.9 to 16.4 t C-$CO_2$ ha$^{-1}$ a$^{-1}$. They also found that management is a main influencer of TER, where intensively managed grasslands produce higher emissions than extensively managed grasslands. With regard to grazing, we found only a minor direct impact on the measured flux rates (Fig. 7).

Wetland TER: Emissions from the study site G2 vary from 0 to 92 kg C-$CO_2$ ha$^{-1}$ day$^{-1}$ and are higher than those from G1, especially in the growing season. This is due to the higher above ground biomass of the different species present and represents a common pattern in unmanaged grasslands (Soussana et al., 2007). Emissions typically end with the cessation of pasture growth during temperatures under 5°C (Parsons, 1988). The annual emissions are 12.54 t C-$CO_2$ ha$^{-1}$ a$^{-1}$, driven by the growing season.

Forest below ground respiration: The mean measured belowground respiration spans between minimum values of 2.1 to 4.5 and maximum values of 9.3 to 19.9 kg C-$CO_2$ ha$^{-1}$ day$^{-1}$ between the different transects (W1-3). While we found higher emissions in the summer months, seasonal differences have a lower magnitude of TER on arable and grassland. This was expected, as we do not measure above ground biomass respiration on our forest study site. Overall, rewetting has the strongest influence on changes in belowground respiration in our forest study sites. The highest emissions occurred in July 2014 after several rewetting events of the uppermost soil layer (Fig. A1). Xiang et al. (2008) reported that multiple rewetting leads to respiration rates of up to eight-times higher. The total yearly soil emissions are 2.98 $\pm$ 0.89 t C-$CO_2$ ha$^{-1}$ a$^{-1}$, which is at the lower end of other European forest ecosystems, e.g., 6.6 $\pm$ 2.9 t C-$CO_2$ ha$^{-1}$ a$^{-1}$, as reported by Janssens et al., (2001). The uphill transect W1 has the highest emission rates throughout the year and shows significant differences when compared to W2 and W3. This transect is less shaded by trees, resulting in a 1.3°C higher annual mean soil temperature compared to W2 and W3, likely causing higher $CO_2$-emissions (Table 3).

**Table 3: Mean measured annual fluxes (Nov 2013 - Dec 2015) from the different land use transects of the Vollnkirchener Bach study area. Differences between the investigated transects and land uses for measured and modelled $CO_2$ emissions in t C-$CO_2$ ha$^{-1}$ a$^{-1}$. * = significant difference (p < 0.05, Kruskal-Wallis test). Arable (A1-3), Grassland (G1), Wetland (G2), Forest (W1-3), RMSE in kg C-$CO_2$ ha$^{-1}$ day$^{-1}$.**

| | A1 | A2 | A3 | G1 | G2 | W1 | W2 | Measured | Mean measured | Mean simulated | Posterior RMSE |
|---|---|---|---|---|---|---|---|---|---|---|---|
| A1 | | | | | | | | 20.10 | 19.96 | 20.53 | 30.73 - 36.38 |
| A2 | | | | | | | | 22.25 | | | 35.66 - 42.26 |
| A3 | | | | | | | | 17.54 | | | 22.90 - 28.46 |
| G1 | | | | | | | | 11.79 | 11.79 | 13.24 | 7.01 - 9.08 |
| G2 | | | | | | | | 12.54 | 12.54 | - | not simulated |
| W1 | * | * | * | * | * | | | 4.00 | 2.98 | 3.28 | 3.53 - 3.89 |
| W2 | * | * | * | * | * | * | | 2.38 | | | 3.37 - 4.07 |
| W3 | * | * | * | * | * | * | | 2.56 | | | 3.15 - 3.96 |

## 3.3 Modelled N fluxes

After selecting the posterior model runs as described in section 2.3.2, we found the model to be generally capable of reproducing the measured data and consequently investigated the modelled C and N cycles in more detail. The modelled $N_2O$ emissions are shown for the different land uses over time (Fig. 2), separated into different seasons (Fig. 3) and before/after management-events occur (Fig. 4). The complete modelled N cycle is given in Table 4.

Arable land N cycle: The arable land simulations consider an annual N input of 198 kg N ha$^{-1}$ a$^{-1}$. This input is balanced by 108.6 ± 50.1 kg N ha$^{-1}$ a$^{-1}$ gaseous (primarily N$_2$), 30.0 ± 29.9 kg N ha$^{-1}$ a$^{-1}$ nitrate leaching and 99.7 ± 7.8 kg N ha$^{-1}$ a$^{-1}$ harvest losses (Table 4), meaning that the modelled outputs are higher than the given inputs. This gap in the annual N cycle is fed by soil storage in the model, indicating N depletion over time. Even though N losses through NO$_3^-$ and particularly N$_2$O emissions

(7.3 ± 2.3 kg N ha$^{-1}$ a$^{-1}$) are only a minor proportion of the total N balance, both rates are high regarding their environmental impacts as a GHG contributing to global warming and as a water pollutant regarding eutrophication and drinking water supply, respectively. However, the uncertainty related to our estimated NO$_3^-$ leaching rate is overall the largest source of uncertainty in our N balance. These estimates cannot be sufficiently constrained with the given observation data, but they are in accordance with other reported N leaching rates on arable land in Germany (Siemens and Kaupenjohann, 2002).

The simulated N$_2$O emissions contribute 3.1% to the total simulated N losses. The underlying model runs follow the trend of the observation data. Hot moments can be observed after fertilizer applications, and they are predicted by the model in time but sometimes not in magnitude (e.g., March to May 2014). During these events, soil moisture is often not modelled accurately: The model predicts rewetting processes that have not been measured at the same magnitude (Fig. A1), which might explain the overestimated fluxes. One possible reason may also be uncertain rainfall model input data. Kavetski et al. (2006) found the

measurements of precipitation within a catchment to be uncertain, as the trajectory of storm cells through a catchment may be different for each storm and may not have their centres at the rain gauge, where rainfall inputs are traditionally measured. Our rainfall data are measured 4 km northeast of the trace gas study area and is likely affected by such uncertainties.

The total simulated and measured emissions on the arable site are highest in the spring (Fig. 3). While the transects A1 and A2 vary, with 95% of the values between 0 and 0.05 kg N$_2$O-N ha$^{-1}$ day$^{-1}$, A3 shows more variation, up to

0.15 kg N$_2$O-N ha$^{-1}$ day$^{-1}$. As A3 is located at the hill toe, we attribute this effect to the lateral transport of nitrate from uphill. However, our one-dimensional model setup does not cover lateral water and nutrient transport; accordingly, the model is not able to predict the higher emissions at A3 in the spring. While such a process is part of complex integrated hydro-biogeochemical catchment models (Haas et al., 2013; Klatt et al., 2017; Wlotzka et al., 2013), it has not yet been confirmed experimentally. The distributions of the measured emissions in the summer, autumn and winter seasons are well in accordance

with the modelled emissions. Furthermore, the modelled emissions are also in agreement with emissions measured before and after manure applications (Fig. 4). This result agrees with a study by Molina-Herrera et al., (2016) who found LandscapeDNDC to be capable of simulating agricultural N$_2$O emissions. However, in our case, the model overestimates peak emissions before fertilizer applications, which leads to higher mean annual modelled emissions (7.33 kg N$_2$O-N ha$^{-1}$ a$^{-1}$). This is 2.8 kg N$_2$O-N ha$^{-1}$ a$^{-1}$ higher than our observed emissions and is even outside the large model uncertainty of

2.3 kg N$_2$O-N ha$^{-1}$ a$^{-1}$. Hence, future research should specifically investigate the reason for this overestimation of peaks, either by revising the model structure or by identifying other sources of model uncertainty.

Grassland N cycle: Grassland simulations consider an annual N input of 12.7 kg, with 7.6 kg coming from modelled biomass that is transferred into dung and urine applied by grazing sheep. The simulated N loss is substantially larger than the N input, with 22.3 ± 13.3 kg N ha$^{-1}$ a$^{-1}$ gaseous losses (primarily N$_2$), 1.5 ± 3.19 kg N ha$^{-1}$ a$^{-1}$ occurring as nitrate leaching and

$29.8 \pm 9.4$ kg N ha$^{-1}$ a$^{-1}$ as biomass removal through grazing sheep and harvest (hay making). Comparing inputs and outputs, we simulated a mean nitrogen gap of $40.9 \pm 25.9$ kg N ha$^{-1}$ a$^{-1}$. The model suggests decreasing soil organic N stocks. So far, we have only initial measurements of soil organic N content. However, we assume that the source of additional N in the form of nitrate in shallow groundwater is a potential dominating process that is not included in the current LandscapeDNDC version

we used. Liebermann et al. (2017) used a revised LandscapeDNDC setup for hypothesis testing to identify potential additional N sources in groundwater-dominated grasslands and showed that groundwater N uptake is a likely contributor.

Taking a closer look at the modelled N$_2$O emissions, one can see that the model did not reproduce high or negative (N$_2$O uptake) emissions. Currently, LandscapeDNDC does not consider any N$_2$O uptake, and accordingly, negative fluxes cannot be simulated by the model. The peaky dynamics of the simulated N$_2$O emissions, especially from August 2014 to January

2015, are not confirmed by the measurements, indicating possible measurement errors during this period of time. In a grazed system with, in our case, approximately 70 sheep per hectare, the animal urine patches create emissions hot spots. With only five chambers, it is possible that the measurements could miss these hot spots. Additionally, the LandscapeDNDC model will assume that the manure is uniformly spread over the field, producing emissions that are likely to be higher than those from non-urine patches, but lower than those from urine patches. One has also to consider the temporal mismatch of our weekly

N$_2$O measurements and the hourly simulations, making a full match of the observations with the simulations difficult. So far, there is no clear effect of grazing on the N$_2$O emissions on the grassland site in both the measurements and modelled results (Fig. 4). The mean modelled annual emissions overestimate the observations by 0.4 kg N$_2$O-N ha$^{-1}$ a$^{-1}$, and even the simulated uncertainty bounds of 0.27 kg N$_2$O-N ha$^{-1}$ a$^{-1}$ do not capture the measured dynamics.

Forest N cycle: The N input is given for the forest model only considering atmospheric deposition with an annual amount of

5.1 kg N ha$^{-1}$ a$^{-1}$. Gaseous losses amount to $1.8 \pm 2.0$ kg N ha$^{-1}$ a$^{-1}$. Leaching contributes to 2.0% of the N output. The rest ($3.3 \pm 2.0$ kg N ha$^{-1}$ a$^{-1}$) is allocated into biomass and soil. By taking a closer look at the N$_2$O emissions (Fig. 2), we see that the model fails to reproduce the observed emission dynamics. The observed N$_2$O emissions have high error bars, and not all transects are driven by frost-thaw cycles or N$_2$O uptake at the same time (Table 2). Parameterizing and simulating the forest transects independently from each other would improve the simulations. One limiting factor is that both N$_2$O uptake and frost-

thaw cycles are not included in the current version of LandscapeDNDC. We therefore recommend the inclusion of frost-thaw cycles (e.g., based on De Bruijn et al., 2009) in the model, as this process can have a major influence on N$_2$O inventories, e.g., up to 73% of the total annual N$_2$O loss at a forest site in Högelwald, Germany (Papen and Butterbach–Bahl, 1999). The mean modelled annual emissions ($0.33 \pm 0.15$ kg N ha$^{-1}$ a$^{-1}$) overestimate the observed emissions on all transects.

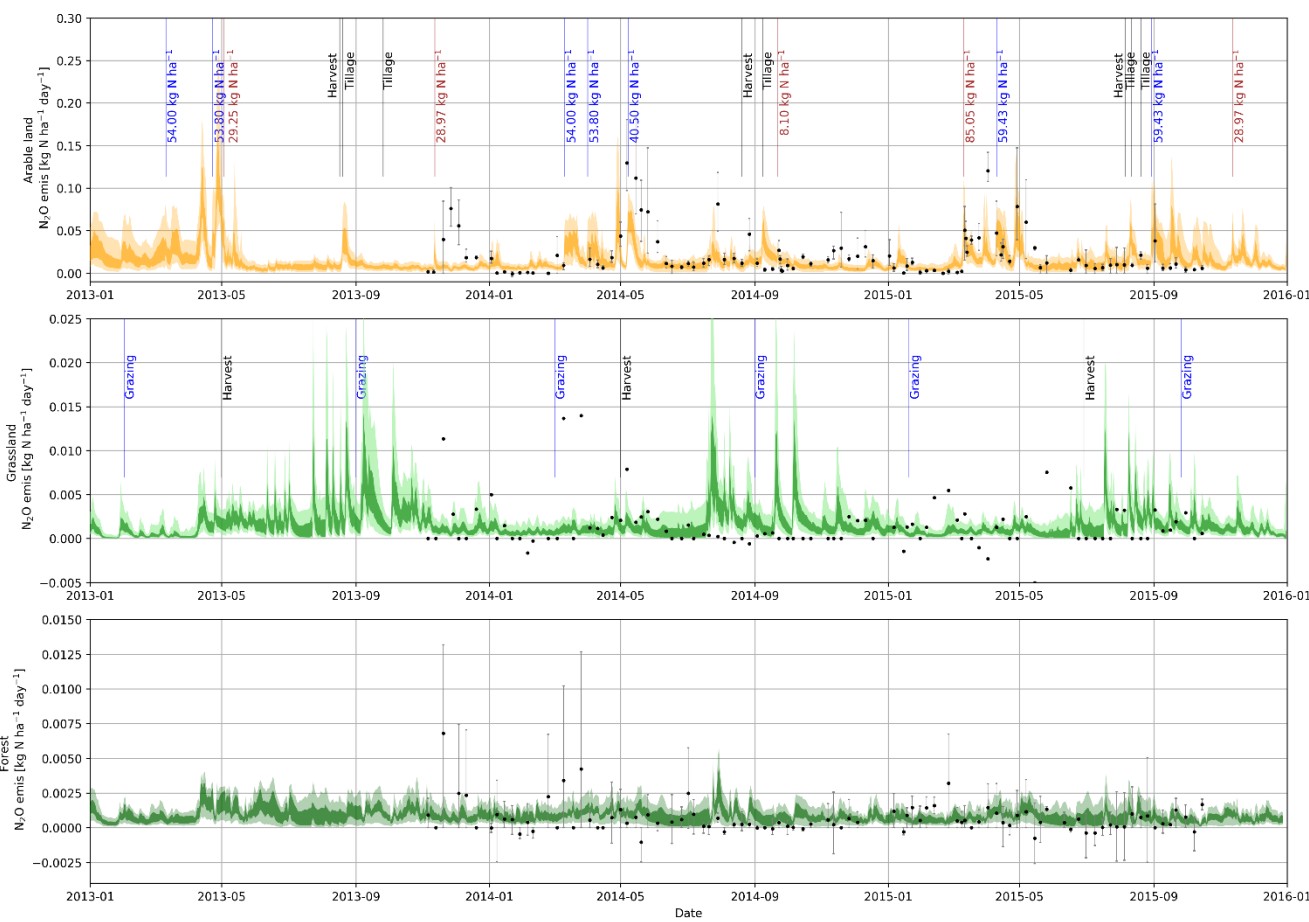

**Figure 2: Measured and modelled N₂O emissions from different land use. Measurements are given as grey error bars showing the variance between the replicated transects and the mean value as a black dot. Posterior model uncertainty is given in light colour for the 5 and 95 percentiles and dark colour for the 25 and 75 percentiles. Vertical lines indicate management events. In the uppermost panel, blue coloured vertical bars indicate fertilizer application, while brown colours indicate manure application.**

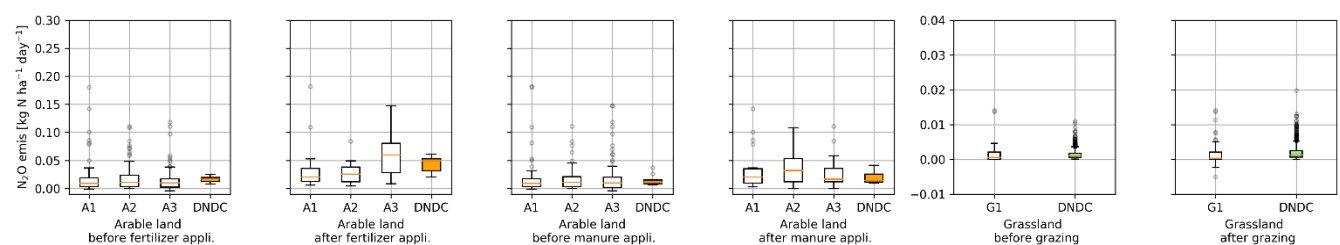

**Figure 3: Observed and modelled N₂O emissions for spring (21ˢᵗ Mar. - 20ᵗʰ Jun.), summer (21ˢᵗ Jun. - 20ᵗʰ Sep.), autumn (21ˢᵗ Sep. - 20ᵗʰ Dec.), and winter (21ˢᵗ Dec. - 20ᵗʰ Mar.).**

**Figure 4: Management effects on N₂O emissions. Measured and modelled emissions within a time window of 2 weeks before and 2 weeks after a management.**

**Table 4: Simulated nitrogen fluxes given by posterior model runs and their uncertainty on different land use in [kg N ha$^{-1}$ a$^{-1}$]. N manure on grassland includes urine and dung input by sheep. Biomass output on grasslands combines harvest export and biomass leaving the system through sheep. Arable land model assumes 20% return of stubble to field.**

| Modeled N flux | Arable land | | Grassland | | Forest | |
|---|---|---|---|---|---|---|
| N deposition | 5.11 | | 5.11 | | 5.11 | |
| N manure | 57.55 | | 7.57 | | 0 | |
| N fertilizer | 135.37 | | 0 | | 0 | |
| **Total input** | **198.03** | | **12.68** | | **5.11** | |
| NO emis. | 0.57 | ±0.16 | 0.46 | ±0.21 | 0.45 | ±0.33 |
| N$_2$ emis. | 62.55 | ±26.83 | 18.69 | ±10.91 | 1 | ±1.5 |
| N$_2$O emis. | 7.33 | ±2.3 | 0.69 | ±0.27 | 0.33 | ±0.15 |
| NH$_3$ emis. | 38.15 | ±20.8 | 2.45 | ±1.89 | <0.01 | ±<0.01 |
| **Total gaseous output** | **108.6** | **±50.09** | **22.29** | **±13.28** | **1.78** | **±1.98** |
| DON leaching | 0.01 | ±<0.01 | 0.01 | ±<0.01 | 0.01 | ±<0.01 |
| NO$_3$ leaching | 30.01 | ±29.9 | 1.46 | ±3.19 | 0.03 | ±0.04 |
| **Total leaching output** | **30.02** | **±29.9** | **1.47** | **±3.19** | **0.04** | **±0.04** |
| N grain export | 63.92 | ±5.17 | 0 | | 0 | |
| N straw export | 35.75 | ±2.67 | 29.77 | ±9.44 | 0 | |
| **Total biomass output** | **99.67** | **±7.84** | **29.77** | **±9.44** | **0** | |
| **Balance** | **-40.26** | **±87.83** | **-40.85** | **±25.91** | **3.29** | **±2.02** |

## 3.4 Modelled C fluxes

The modelled $CO_2$ emissions are shown for the different land uses over time (Fig. 5), separated into different seasons (Fig. 6) and before/after management events (Fig. 7). The complete modelled C cycle is given in Table 5.

Arable land C cycle: The LandscapeDNDC simulations for the arable system predict a mean annual gross carbon uptake of
10   $25.7 \pm 1.3$ t C-$CO_2$ ha$^{-1}$ a$^{-1}$. $20.5 \pm 1.8$ t C-$CO_2$ ha$^{-1}$ a$^{-1}$ leaves the system through respiration, to which maintenance respiration contributes the largest proportion (65%). This is in accordance with annual measured losses (Table 3). The harvest output is with $4.7 \pm 0.4$ t C ha$^{-1}$ a$^{-1}$ and is in good agreement with the observed yields (Fig. A4). However, the temporal dynamics of the modelled TER on the arable land study site underestimate the emissions in the summer season (Fig. 6), and the mean modelled fluxes are substantially lower than those measured before and after the harvest (Fig. 7).
15   Tillage and harvest events occur in the summer season. While the observed emissions drop after harvest by 25%, the modelled emissions drop by 50%. The reason for this is either an underestimation of the emissions through LandscapeDNDC (after harvest events until tillage occurs) or uncertainties in the measured $CO_2$ emissions upscaling method (discussed in chapter 2.1).

As microbial processes can oxidize more soil carbon after harvests (resulting in higher heterotrophic respiration), we assume that the discrepancy stems from the model simulations. There are studies, e.g., Buyanovsky et al. (1986), which report the highest soil respiration rates after harvests. The modelled and measured soil $CO_2$ emissions agree well after tillage. However, unless there is a gap of two weeks or more between harvest and tillage, the "pre-tillage" results will include some post-harvest

effects, and the "post-harvest" results will also include some post-tillage effects. Our intention to present the data grouped by these events are the discrepancies between modeled and observed $CO_2$ dynamics. There is a sharp drop of modeled $CO_2$ emissions after harvest due to the prompt absence of autotrophic respiration. In reality, there will likely be some ongoing metabolic respiration of plant tissue remaining in the field, which is not represented by the 'assumed' dead plant material in the model. After incorporation of harvest residues (at tilling) modeled $CO_2$ emissions increase again sharply. The sharp increase

is due to the incorporation and hence availability of fresh litter (stubble) and a temporary stimulation of decomposition by the model due to the disruption/aeration of the soil structure. Both, overestimation of fresh litter and/or stimulation of decomposition by the model may contribute to the discrepancies between observed and modelled $CO_2$ emissions.

Grassland C cycle: The LandscapeDNDC simulations for the grassland system (G1) predict a mean annual gross carbon uptake of $16.9 \pm 1.7$ t C-$CO_2$ ha$^{-1}$ a$^{-1}$ and an annual loss of $13.2 \pm 2.3$ t C-$CO_2$ ha$^{-1}$ a$^{-1}$ through respiration. The rest is related to grazing

($0.2 \pm< 0.01$ t C-$CO_2$ ha$^{-1}$ a$^{-1}$), harvesting ($2.1 \pm 0.7$ t C-$CO_2$ ha$^{-1}$ a$^{-1}$) and allocation in the soil ($1.4 \pm 4.7$ t C-$CO_2$ ha$^{-1}$ a$^{-1}$). The model cannot determine whether the system is net gaining or losing carbon. The annual mean and temporal dynamics of the modelled emissions are well in accordance with the measured emissions. The effect of grazing has a minor influence on the total ecosystem respiration (Fig. 7), resulting in a wider range of both measured and modelled emissions. Grazing, i.e., the reduction of root biomass, results in two contrary processes: a reduction in maintenance respiration and an increase in

autotrophic respiration (Raich and Tufekciogul, 2000).

Forest C cycle: The forest model predicts an annual C input of $8.9 \pm 0.6$ t C-$CO_2$ ha$^{-1}$ a$^{-1}$, which is quite low compared to the estimations for old-growth beech forests in Europe, with reported rates from 14.4 to 18.3 t C-$CO_2$ ha$^{-1}$ a$^{-1}$ (Molina-Herrera et al., 2015). However, C uptake rates vary in magnitude, with values ranging from 3 to 34 t C-$CO_2$ ha$^{-1}$ a$^{-1}$ for different forests in different growing stages (Waring et al., 1998). As our study site is a mixture of young and old beech trees, we assume that

it has 40 - 50% less biomass compared to an old beech forest. Of the modelled C input, $6.6 \pm 0.5$ t C-$CO_2$ ha$^{-1}$ a$^{-1}$ leaves the system as gaseous $CO_2$. The rest is accumulated in the biomass and soil. The annual mean and dynamics of the modelled emissions are in accordance with the measured emissions. We expected to see rising emissions with litter fall in autumn (Raich and Tufekciogul, 2000), but cannot report this effect, either with measurements or with model results (Fig. 6).

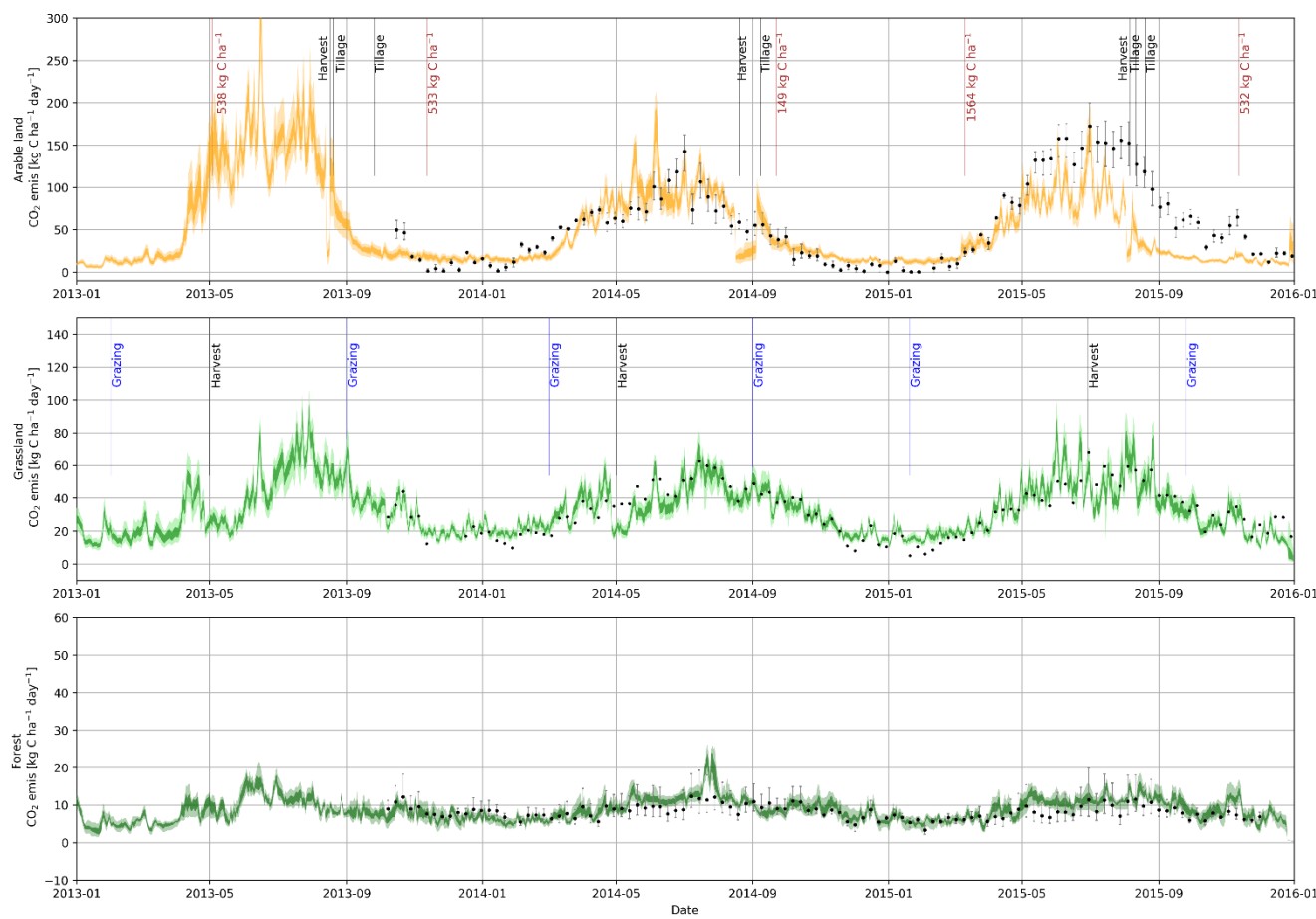

**Figure 5: Modelled CO₂ emissions and management.** Measurements are given as grey error bars showing the variance between the replicated transects and the mean value as a black dot. Posterior model uncertainty is given in light colour for the 5 and 95 percentiles and dark colour for the 25 and 75 percentiles. Vertical lines indicate management events. Brown coloured bars in the uppermost panel indicate manure application.

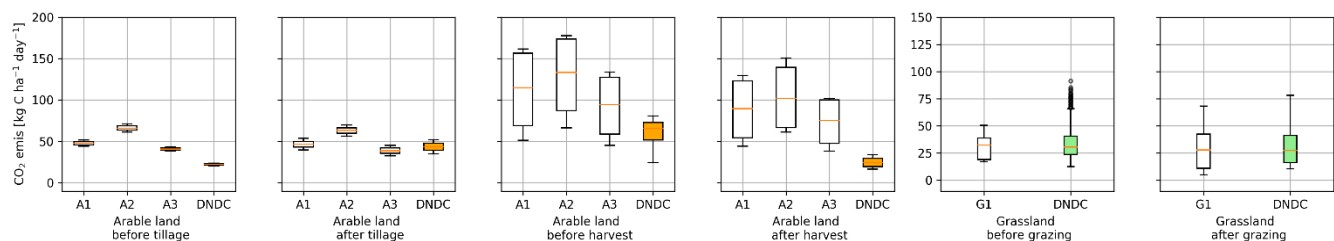

**Figure 6: Observed and modelled $CO_2$ emissions for spring (21st Mar. - 20th Jun.), summer (21st Jun. - 20th Sep.), autumn (21st Sep. - 20th Dec.), and winter (21st Dec. - 20th Mar.).**

**Figure 7: Management effects on $CO_2$ emissions. Measured and modelled emissions in a time window of 2 weeks before and 2 weeks after a management.**

**Table 5: Simulated carbon fluxes given by posterior model runs and their uncertainty on different land use in [t C ha$^{-1}$ a$^{-1}$]. C manure on grassland includes input by sheep's dung. Arable land model assumes 20% return of stubble to field.**

| Modeled C flux | Arable land | | Grassland | | Forest | |
|---|---|---|---|---|---|---|
| CO$_2$ uptake | 24.65 | ±1.32 | 16.8 | ±1.72 | 8.94 | ±0.56 |
| C manure | 1.06 | | 0.07 | | 0 | |
| **Total input** | **25.71** | **±1.32** | **16.87** | **±1.72** | **8.94** | **±0.56** |
| Growth respiration | 2.53 | ±0.2 | 0.81 | ±0.27 | 1.44 | ±0.05 |
| Heterotrophic respiration | 4.69 | ±0.53 | 2.27 | ±0.9 | 2.04 | ±0.1 |
| Maintenance respiration | 13.31 | ±1.06 | 10.16 | ±1.13 | 3.11 | ±0.39 |
| **Total gaseous output** | **20.53** | **±1.79** | **13.24** | **±2.3** | **6.59** | **±0.54** |
| DOC leaching | <0.01 | ±<0.01 | <0.01 | ±<0.01 | <0.01 | ±<0.01 |
| **Total leaching output** | **<0.01** | | **<0.01** | | **<0.01** | |
| C bud export | 1.97 | ±0.17 | 0 | | 0 | |
| C straw export | 2.75 | ±0.21 | 2.28 | ±0.72 | 0 | |
| **Total biomass output** | **4.72** | **±0.38** | **2.28** | **±0.72** | **0** | |
| **Balance** | **0.46** | **±3.49** | **1.35** | **±4.74** | **2.35** | **±1.1** |

## 4 Conclusion

We presented a two-year measurement campaign of trace gas emissions from adjacent land uses i.e., arable land, grassland and forest ecosystems, with concurrent model development and rigorous testing through a model-data fusion.

We found high emissions of N$_2$O and CO$_2$ on our arable land sites, low emissions on grassland sites and the lowest emissions on the forest sites. These observations enable us to investigate the underlying effects of plant growth, temperature and WFPS, land use effects, seasonal patterns and management effects. Respiration amounts rise in less shaded (warmer) areas of the forest, while N$_2$O emissions increase towards the foothills of the forest and arable land sites due to nitrogen accumulation. Highly variable N$_2$O emissions in forests resulted in large uncertainties in the model verification data, which translated into large uncertainties in the model results for forests.

Detailed measured data on soil and management allowed us to fit the biogeochemical model LandscapeDNDC to the measured soil moisture, yield and GHG emissions of CO$_2$ and N$_2$O. A subjective conclusion about the overall model performance is shown in Table 6: The model reproduced the measured data reasonably well in time, separated into seasons and management events. The model performance was best in predicting management effects on N$_2$O emissions and annual CO$_2$ emissions for all land uses. With regard to land use, the simulations for grassland sites work best, followed by those for arable land. The simulations for N$_2$O emissions on arable land outperform those for CO$_2$, and vice versa for grassland. Low emissions on forest sites were generally difficult to depict using our modelling approach.

**Table 6: Overall posterior model performance of LandscapeDNDC on different land uses in reproducing GHG emission data. Subjectively classified into (1) good, (2) medium and (3) poor model performance in simulating reliable annual sums, seasonal patterns and magnitudes of management events (e.g., fertilizer application). NA = not applicable, i.e., no forest management during modelled period from 2010-2016.**

| Modelled performance on each land use | $N_2O$ emissions | | | $CO_2$ emissions | | |
|---|---|---|---|---|---|---|
| | annual | seasonal | management | annual | seasonal | management |
| Arable land (A1-3) | 2 | 1 | 1 | 1 | 2 | 3 |
| Grassland (G1) | 1 | 2 | 1 | 1 | 1 | 1 |
| Forest (W1-W3) | 2 | 2 | NA | 1 | 2 | NA |

The model-data fusion approach allowed us to identify model structural deficiencies that would likely increase model performances if addressed in Landscape DNDC: missing $N_2O$ uptake processes; missing $NO_3^-$ (and potentially dissolved organic nitrogen) uptake through shallow groundwater; missing lateral interaction on hillslopes due to the 1D model setup. Furthermore, posterior model runs allowed for the quantification of the magnitude and uncertainty of unmeasured C and N cycle fluxes. The investigated forest site generally acts as the largest sink for C and N, with annual sequestration rates of $2.4 \pm 1.1$ t C ha$^{-1}$ and $3.3 \pm 2.0$ kg N ha$^{-1}$. Whether the extensive grazed grassland is also acting as a sink for C with $1.4 \pm 4.7$ t C ha$^{-1}$ per year remains uncertain, while the N cycle of the grassland model cannot be closed with the given settings. Shrinking N soil pools indicate a missing input, which we assume to be shallow groundwater with an additional N supply of approximately $40.9 \pm 25.9$ kg N ha$^{-1}$ a$^{-1}$.

Current land use in this catchment is dominated by forests (37%) and arable land (35%), whereas grassland sites (11%) are mainly distributed along the stream. From the viewpoint of climate-smart landscapes, the measured data suggest the benefit of forests in a landscape, as they have the fewest GHG emissions. Riparian zones can act as sinks of N but only during the vegetation period and during times when roots have access to groundwater. Arable land use produces high amounts of $N_2O$, not throughout the year, but rather, in spring after fertilizer application.

Potential interactions of land use patterns cannot be quantified with the current one-dimensional model approach. However, the dataset could be used in future studies to quantify the nitrate uptake of riparian zones in more detail, e.g., by coupling LandscapeDNDC to a hydrological model, as done by Klatt et al. (2017). Such a model setup would also allow for upscaling in space, e.g., for the generation of GHG inventories or an analysis of more detailed management scenarios in time.

**Code availability**

The LandscapeDNDC framework is free available upon request from www.ldndc.imk-ifu.kit.edu

The SPOTPY tool, used for model-data fusion, is free and open source and is available from www.pypi.python.org/pypi/spotpy

**Data availability**

All measured data are free available upon request from www.fb09-pasig.umwelt.uni-giessen.de:8081

**Appendices**

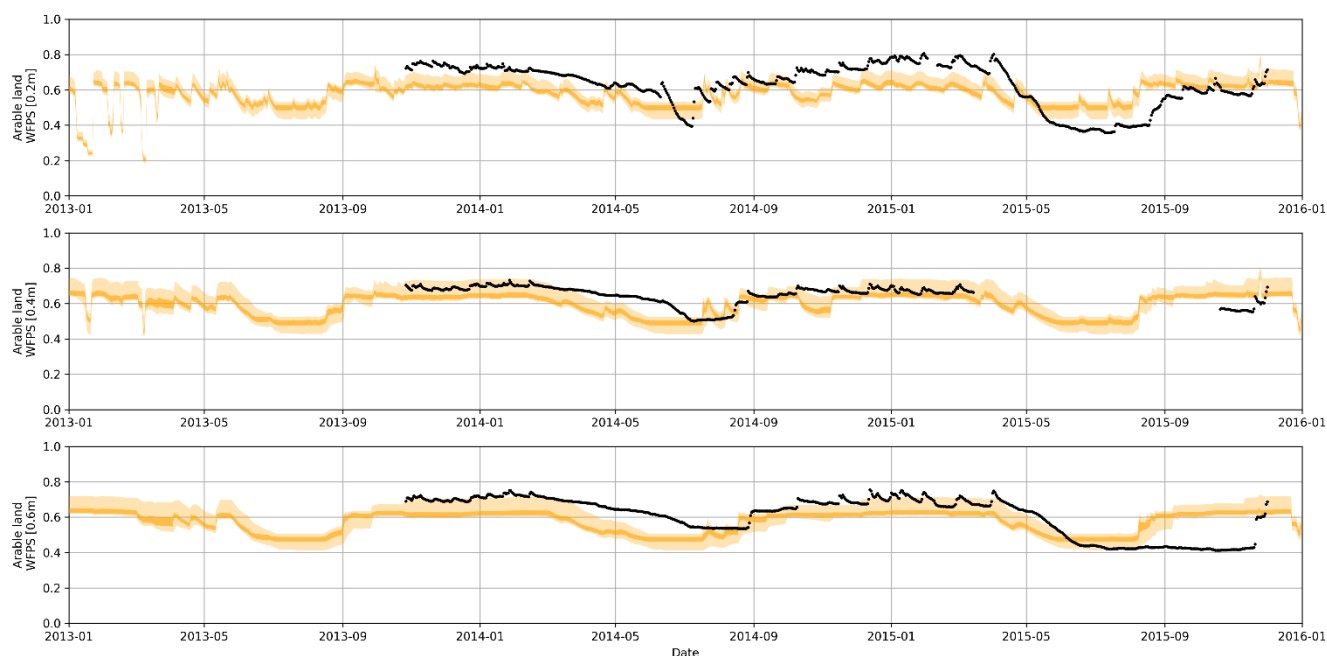

**Figure A1: Modeled WFPS on arable land in different depths. RMSEs ranging from 0.0774 to 0.1194% WFPS [0.2m], 0.0511 to 0.0955% WFPS [0.4m] and 0.0921 to 0.1193% WFPS [0.6m].**

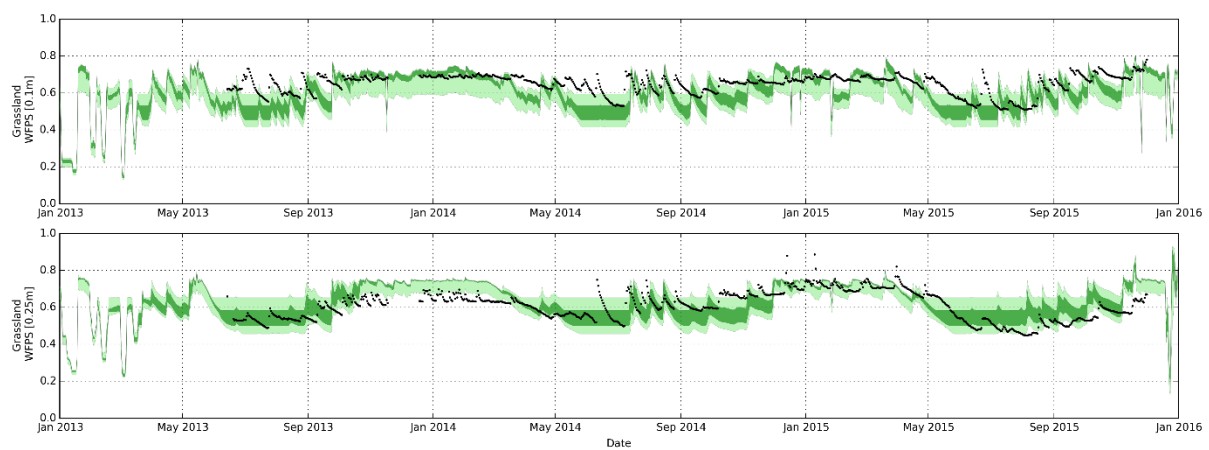

**Figure A2: Modeled WFPS on grassland in different depths. RMSEs ranging from 0.043 to 0.1481% WFPS [0.1m] and 0.056 0.1069% WFPS [0.25m].**

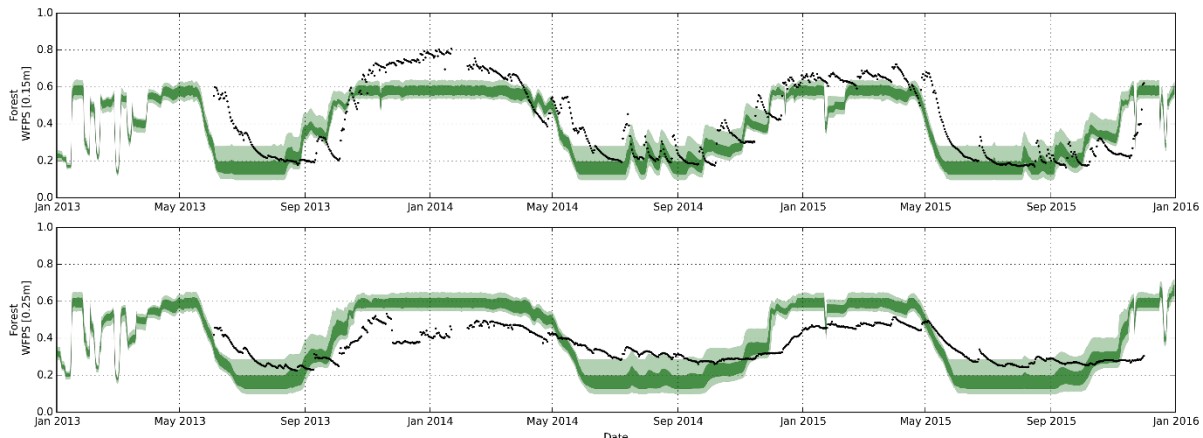

**Figure A3: Modeled WFPS on forest in different depths. RMSEs ranging from 0.0817 to 0.1324% WFPS [0.15m] and 0.0812 to 0.1606% WFPS [0.25m].**

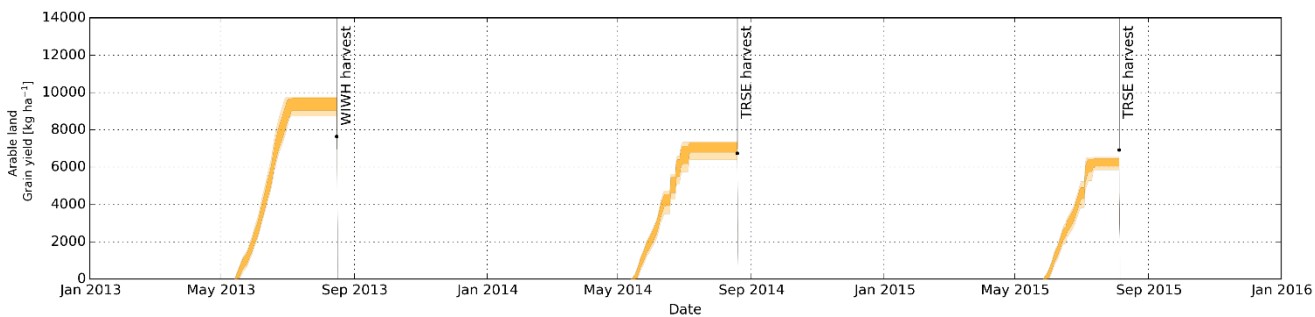

5    **Figure A4: Modeled dry weight grain yield on arable land use. WIWH = Winter wheat, TRSE = Triticum secale. RMSEs ranging from 1125.7 to 2529.2 kg ha⁻¹.**

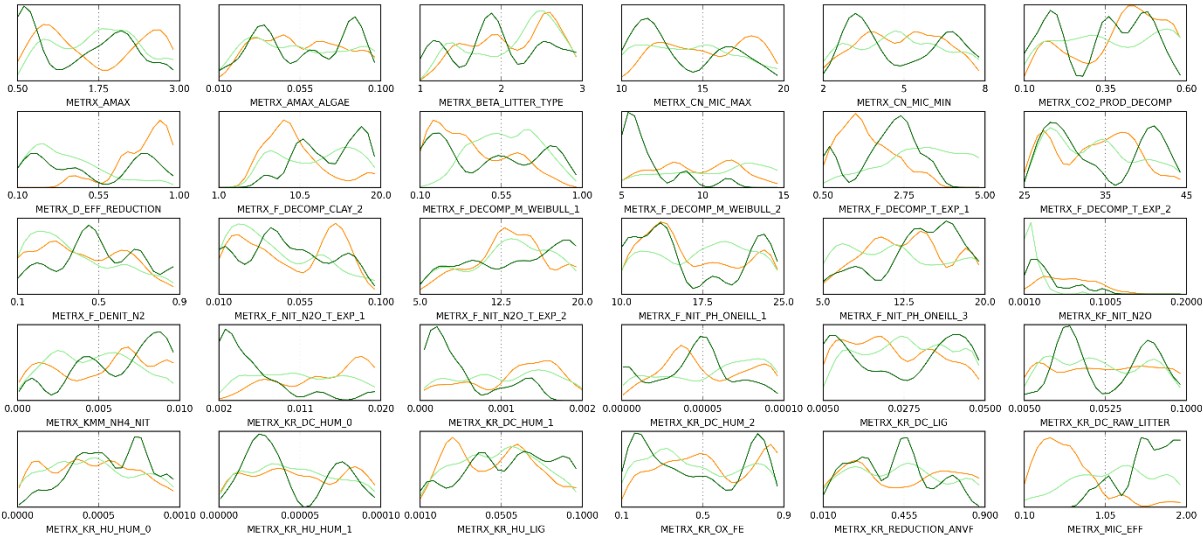

**Figure A5: Posterior parameter distribution of the LandscapeDNDC module MeTrx. Orange line = arable land, light green line = grassland, dark green line = forest model set up.**

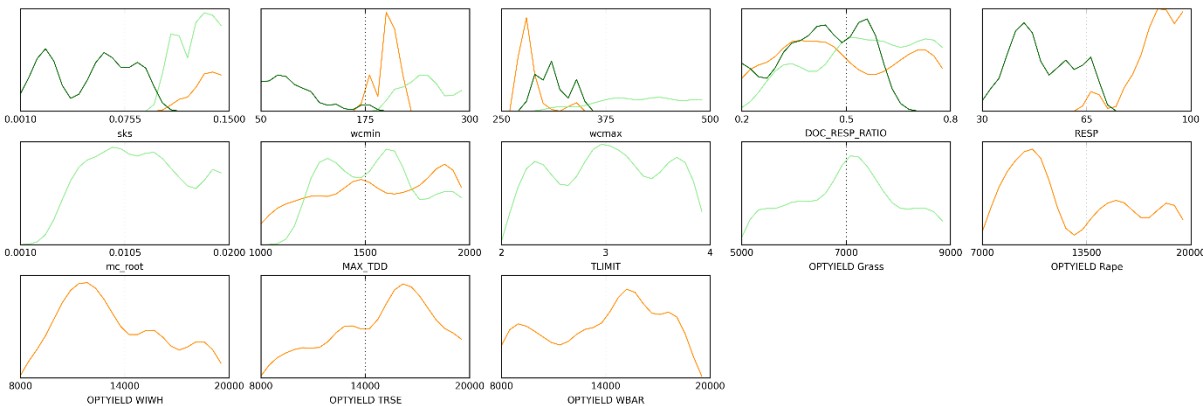

**Figure A6: Posterior parameter distribution of the LandscapeDNDC modules wcDNDC and physiology. Orange line = arable land, light green line = grassland, dark green line = forest model set up.**

**Table A1: Input parameters for all investigated LandscapeDNDC modules with uniform distribution for Latin Hypercube Sampling. FASY = Fagus sylvatica, PERG = Perennial grass.**

| module | parameter name | Description | min | max |
|---|---|---|---|---|
| **wcDNDC** | sks_arable | Value of soil layer for saturated hydraulic conductivity | 0.1 | 0.2 |
| **wcDNDC** | sks_grassland | Value of soil layer for saturated hydraulic conductivity | 0.1 | 0.15 |
| **wcDNDC** | sks_forest | Value of soil layer for saturated hydraulic conductivity | 0.1 | 0.1 |
| **wcDNDC** | wcmin_arable | Wilting point of soil layer | 170 | 220 |
| **wcDNDC** | wcmin_grassland | Wilting point of soil layer | 200 | 300 |
| **wcDNDC** | wcmin_forest | Wilting point of soil layer | 40 | 200 |
| **wcDNDC** | wcmax_arable | Field capacity of uppermost soil layer | 270 | 350 |
| **wcDNDC** | wcmax_grassland | Field capacity of soil layer | 300 | 500 |
| **wcDNDC** | wcmax_forest | Field capacity of soil layer | 270 | 350 |
| **physiology** | DOC_RESP_RATIO_FASY | Ratio of root exudates related to root growth respiration | 0.1 | 0.6 |
| **physiology** | RESP_FASY | Factor determining plant respiration | 30 | 70 |
| **physiology** | DOC_RESP_RATIO_PERG | Ratio of root exudates related to root growth respiration | 0.2 | 0.8 |
| **physiology** | MC_ROOT_PERG | Maintenance respiration coefficient of roots | 0.001 | 0.02 |
| **physiology** | MAX_TDD_PERG | Temperature degree days for full plant development | 1200 | 2000 |
| **physiology** | OPTYIELD_PERG | Optimum yield of crops and grasses | 5000 | 9000 |
| **physiology** | TLIMIT_PERG | Temperature limit for plant growth | 2 | 4 |
| **physiology** | DOC_RESP_RATIO_arable | Ratio of root exudates related to root growth respiration | 0.2 | 0.8 |
| **physiology** | MAX_TDD_arable | Temperature degree days for full plant development | 1000 | 2000 |
| **physiology** | RESP_arable | Factor determining plant respiration | 30 | 100 |
| **physiology** | OPTYIELD_rape | Optimum yield of Rape | 7000 | 20000 |
| **physiology** | OPTYIELD_WIWH | Optimum yield of Winter Wheat | 8000 | 20000 |
| **physiology** | OPTYIELD_TRSE | Optimum yield of Triticale | 8000 | 20000 |
| **physiology** | OPTYIELD_WBAR | Optimum yield of Winter Barley | 8000 | 20000 |
| **METRX** | METRX_AMAX | Maximum microbial death rate | 0.5 | 3 |
| **METRX** | METRX_AMAX_ALGAE | Maximum decay rate of alga | 0.01 | 0.1 |
| **METRX** | METRX_BETA_LITTER_TYPE | Exp. fac. of litter decomposition red. depend. on lignin conc | 1 | 3 |
| **METRX** | METRX_CN_MIC_MAX | Maximum allowed C:N ratio for microbes | 10 | 20 |
| **METRX** | METRX_CN_MIC_MIN | Minimum allowed C:N ratio for microbes | 2 | 8 |
| **METRX** | METRX_CO2_PROD_DECOMP | Instantaneously production of CO2 during decomposition | 0.1 | 0.6 |
| **METRX** | METRX_D_EFF_REDUCTION | Reduction factor for gas diffusion | 0.1 | 1 |
| **METRX** | METRX_F_DECOMP_CLAY_2 | Factor for clay dependency of decomposition | 1 | 20 |

| METRX | METRX_F_DECOMP_M_WEIBULL_1 | Factor for water filled pore space dependency of decomposition | 0.1 | 1 |
|---|---|---|---|---|
| METRX | METRX_F_DECOMP_M_WEIBULL_2 | Factor for water filled pore space dependency of decomposition | 5 | 15 |
| METRX | METRX_F_DECOMP_T_EXP_1 | Factor for temperature dependency of decomposition | 0.5 | 5 |
| METRX | METRX_F_DECOMP_T_EXP_2 | Factor for temperature dependency of decomposition | 25 | 45 |
| METRX | METRX_F_DENIT_N2 | Factor determining amount denitrified nitrogen goes to N2 | 0.1 | 0.9 |
| METRX | METRX_F_NIT_N2O_T_EXP_1 | Factor for temp. depend. of N2O prod. during nitrification | 0.01 | 0.1 |
| METRX | METRX_F_NIT_N2O_T_EXP_2 | Factor for temperature dependency of N2O production | 5 | 20 |
| METRX | METRX_F_NIT_PH_ONEILL_1 | Factor for pH dependency of nitrification | 10 | 25 |
| METRX | METRX_F_NIT_PH_ONEILL_3 | Factor for pH dependency of nitrification | 5 | 20 |
| METRX | METRX_KF_NIT_N2O | Maximum fraction of nitrified NH4 that goes to N2O | 0.001 | 0.2 |
| METRX | METRX_KMM_NH4_NIT | Michaelis-Menten const. for NH4 depend. of nitrification | 0.00001 | 0.01 |
| METRX | METRX_KR_DC_HUM_0 | Decomposition constant of recalcitrant young humus | 0.002 | 0.02 |
| METRX | METRX_KR_DC_HUM_1 | Decomposition constant of recalcitrant old humus | 0.00005 | 0.002 |
| METRX | METRX_KR_DC_HUM_2 | Decomposition constant of recalcitrant old humus | 0.000001 | 0.0001 |
| METRX | METRX_KR_DC_LIG | Decomposition constant of lignin | 0.0005 | 0.05 |
| METRX | METRX_KR_DC_RAW_LITTER | Decomposition constant of raw litter | 0.005 | 0.1 |
| METRX | METRX_KR_HU_HUM_0 | Rate constant for humification of labile humus to recalcitrant young humus | 0.000001 | 0.001 |
| METRX | METRX_KR_HU_HUM_1 | Rate constant for humification of recalcitrant young humus to recalcitrant old humus | 0.000001 | 0.0001 |
| METRX | METRX_KR_HU_LIG | Rate constant for humification of lignin | 0.0001 | 0.1 |
| METRX | METRX_KR_OX_FE | Rate constant of iron oxidation | 0.1 | 0.9 |
| METRX | METRX_KR_REDUCTION_ANVF | Decomposition reduction due anaerobicity | 0.01 | 0.9 |
| METRX | METRX_MIC_EFF | Microbial carbon use efficiency | 0.1 | 2 |

**Author contribution**

T. Houska, L. Breuer and R. Kiese designed and managed the experiments. D. Kraus and T. Houska performed the simulations.
T. Houska prepared the manuscript with contributions from all co-authors.

**Competing interests**

5   The authors declare that they have no conflict of interest.

**Acknowledgements**

We acknowledge the financial support provided by the Deutsche Forschungsgemeinschaft (DFG) for Tobias Houska (BR2238/13-1). Special thanks deserves Felix Kruck, Eva Holthof and Michael Herzog for their fieldwork during any weather conditions, Anja Schaefler-Schmid and Julia Valverde for lab analysis and providing the chamber sampling equipment as well

10  as the farmer, for letting us study his land and providing the detailed management information.

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
