# Peer review of "Constraining a complex biogeochemical model for multi-site greenhouse gas emission simulations by model-data fusion"

_Biogeosciences, 2017_

## Referee Comment (RC1) · Anonymous Referee #1 · 13 Apr 2017

General comments

In this study the authors compare the DNDC model simulations of N2O and CO2 emissions against measurements across three different landuse types (arable cropping, grassland, forest). This comparison against different land-use types, as well as the multi-objective Bayesian model calibration method should be of interest to other researchers in this area.

One weakness of this study is that there is no attempt to verify the model calibrations with a independent data. Given that there were three years' data it would have been possible to use two years for the parameterisation and then test the model using the third year's data. (Although this might not work so well for the arable land where the

crop types change between years).

Another problem is that the CO2-equivalent values of N2O emissions are incorrectly calculated. It should be noted that the GWP values convert from kg N2O to kg CO2, not kg N2O-N to kg CO2-c.

This MS would benefit from editing by a native English speaker as there a numerous grammatical errors. Also, results should be consistently presented with their corresponding errors.

Specific comments

2.2 Trace gas measurements

- Were any measurements other than CO2 and N2O made?

2.3 Modelling approach

- This section contains very little information about how the LandscapeDNDC model works. What processes does it consider, what timestep does it work at, what drivers does it consider? Is it 1, 2, or 3-dimensional? Later in the MS it is mentioned that the model does not consider lateral flows of water and nutrients. This should be mentioned here as many readers might expect that a "Landscape" model would consider horizontal flows.

- pg 5, line 3: If only G1 can be modelled then G2 should not be included in Tables 2 and 3. If G2 is sufficiently different from G1 then it does not make sense to average the results for these two systems

Table 1

- A column with management practices applied (e.g. manure and fertiliser applications) would also be informative

- Why are the ranges for organic C and N expressed as high-low when all the other

ranges are low-high?

3.1 Measured N2O fluxes

- pg 6, line 27: Need to clarify what you mean by "no statistical difference over time". Do you mean no differences in the annual cumulative emissions?

- pg 6, line 27: "Highest emissions occur after management events". This is not always true. For example, between May and Sept 2014 there's a measured peak several months after the last management practice

- pg 6, line 30: CO2e incorrectly calculated. 4.5 kg N2O-N = 7.07 kg N2O (4.5 * 44/28) = 2107 kg CO2e (7.07 * 298) = 575 kg CO2e-C (2107*12/44)

- Grassland N2O: Given that the grazed site and the wetland have different management, soil and vegetation properties it seems strange to combine them both as a single "grassland" type.

- pg 7, lines 21-22: Conversion of emissions to CO2-Ce is incorrect (see comment for pg 6, line 30)

- pg 7, line 22: How was the emission factor of 5.4% calculated? According to Table 4the manure N input was 7.57 kg N/ha/a. Assuming no background emissions this gives an emission factor of 0.29/7.57 *100% = 3.8%

- pg 7, line 7: The percentage differences between the forest transects are incorrectly stated. For example, W1 has 3x the emissions of W2, but this is a difference of +200% not +300%. Similarly for the difference between W3 and W2

- pg 7, line 30: The measured negative fluxes are all small compared to the measurement error. Therefore, how do you know that the negative fluxes are real and not just the result of measurement error?

- pg 7, line 33: Figure A3 shows the WFPS, but it doesn't the correlation between negative emissions and WFPS

- pg 7, line 34: Conversion of emissions to CO2-Ce is incorrect (see comment for pg 6, line 30)

- pg 8, line 1: 0.08 is almost two orders of magnitude smaller than 5.1

Table 2

- The A3:A3 and G1:G1 squares should be blocked out

- G1 and G2 should be separated into two separate categories (grassland and wetland)

3.2 Measured CO2 fluxes

- I disagree with the decision to use different definitions of 'CO2 emission' depending on land use. I think it would be clearer to use distinct terms such as TER and below-ground respiration to avoid the potential for reader confusion

- In general measured values should be quoted with uncertainties in this section

- pg 8, line 21-22: It is confusing to talk about a weekly measured value and then give the units in $day^{-1}$

- pg 8, line 21-22: What are these results "not significantly different" from?

- pg 9, line 5: What were the measured CO2 fluxes negatively correlated with?

Table 3

- The measurements from G1 and G2 should not be averaged as these systems were sufficiently different that only G1 was able to be modelled. Also, Table 2 explicitly states that G2 was not modelled. This should also be stated in Table 3

Figures 3 and 6

- For the arable and grassland sites the total DNDC results contain values higher than any that occur in the individual seasons. How is this possible?

- It is confusing to include G2 in the grassland results as the model was only run for the

G1 system

3.3 Modeled N fluxes

- pg 10, line 17: The uncertainty in NO3- leaching is actually the largest in Table 4

- pg 11, line 1: Need to be more specific about what you mean when you say emissions were highest in spring. Are you talking about the total, mean, median, variance, upper quartile, or maximum of the emissions?

- pg 11, line 16: It is not clear what is the simulated N loss large in comparison to? It could be the grassland N inputs or the losses from the arable system.

- pg 11, line 19: Do you have any evidence of what is happening to organic N stock in the real system?

- pg 11, 19-20: It does not make sense to say that the model is mimicking an additional N source that is not included in the model. The model can only simulate what has been included in the model.

- pg 11, line 26-29: What stocking rate was used for the grazing? Note that for grazed systems the emissions will be spatially as well as temporally peaky. In the grazed system the animal urine patches will create emissions hot spots. With only 5 chambers it is possible that the measurements could miss these hot spots. Meanwhile, the DNDC model will assume that the manure is uniformly spread over the field, producing emissions that are likely to be higher than those from non-urine patches, but lower than those from urine patches.

- pg 11, line 34: Total output is 1.82, total leaching is 0.04, therefore leaching is 0.04/1.82 *100% = 2% of output.

- pg 12, line 7-9: Measured range 0.18-0.48 kg N2O-N/ha/a does not overlap the measured range of 0.03-0.09 kg N2O-N/ha/a.

Table 4

- Should the forest NH3 emission be "<0.01" rather than ">0.01"?

3.4 Modeled C flux

- pg 14, line 12-13: Not sure what the relevance of Figure 7 is here. The statement "mean modeled fluxes are substantially lower than measured ones" contradicts the results in Table 3

- Arable C cycle: There will be some confounding effects in the before/after tillage and before/after harvest emissions in Figure 7. Unless there is a >2 week gap between harvest and tillage the "pre-tillage" results will include some post-harvest effects and the "post-harvest" results will also include some post-tillage effects. Some discussion of how the model handles tillage and harvest events might be informative here.

- pg 15, line 6: it is a little odd to describe increasing soil C as an "output" as the C is remaining within the system

- pg 15, line 6: Include uncertainties here. In particular it is important to note that the model cannot determine whether the system is net gaining or losing carbon.

Table 5

- Should DOC leaching be "<0.01" rather than ">0.01"?

Conclusion

- Table 6: New results shouldn't be presented in the conclusions. This Table should be in the Results and discussion. There also needs to be an explanation of how the model performance was classified as good, medium, or poor.

- Include uncertainties with results

- It is uncertain whether the grassland was acting as a sink or source of C as the balance 1.35 +/- 4.74 t C/ha/a. Therefore both positive and negative values are within the uncertainty range.
Technical corrections

pg 2, line 14-15: Models are not "driven by" uncertainties. Might be better to use "with"?

pg 2, line 15-16: Revise to "During model application further uncertainties arise due to the uncertainties in the applied forcing data"

pg 6, line 24-25: reference to (Fig. 3), (Fig. 4), and (Fig. 5) should be (Fig. 2), (Fig. 3), and (Fig. 4) respectively

pg 6, line 25: Table 4 is referred to before Table 3

pg 7, line 28: "contribute" should be "attribute"

pg 8, line 20: "N" should be "C"

pg 14, line 10: Delete "perfect"

pg 14, line 17: There is no section 3.1.2

––––––––––––––––––––––––––––––––––––––

---

## Referee Comment (RC2) · Anonymous Referee #2 · 21 Apr 2017

The authors used a model-data fusion method to constrain the LandscapeDNDC model with multiple-year greenhouse gas fluxes from arable, forest and grassland sites. I appreciate the authors' effort to parameterize biogeochemistry models using long-term field data. I would like the authors to further clarify some of my concerns: (1) Is LandscapeDNDC a multi-layer (i.e., vertically-resolved) model? If yes, how many soil layers are included and how do you model the fluxes between layers? (2) In Tier I and Tier II, do you include the same parameters, e.g., all parameters in Table A1? To my understanding, hydrological parameters (e.g., in wcDNDC module) controlling soil water (represented by WFPS) in Teir I will also influence N2O and CO2 in Teir II; however, some key parameters governing the biogeochemistry processes (e.g., in METRX module) do not necessarily affect soil water. (3) page 6: please further explain "within best 5% of all simulated RMSEs". Do you mean the best 5% of total number of simulations or the best 5% of unduplicated RMSEs? If it's the former, the number of accepted model runs depends on the total number of model runs. Since multiple objectives (e.g., WFPS in different depths) are considered, do you integrate them (RMSEs) into one single objective? If not, how do you determine the acceptance of a model run? (4) Table 6: what are the criteria used to classify model performance? Other minor comments: (5) Table 1: Since the model DNDC include inorganic N, what's the inorganic N amount used in the model? How do you initialize the model? Using measured values (e.g., Table 1) or implementing model spin-up? (6) please use month names (e.g., January, February, . . .) to indicate the month in a date (e.g., 01.11.2013)

---

## Author Comment (AC1) · 7 Jun 2017

(comments of the referees are printed in blue, responses of authors are held in black)

We would like to thank the reviewer #1 for his highly constructive comments on the manuscript bg-2017-96 "Constraining a complex biogeochemical model for multi-site greenhouse gas emission simulations by model-data fusion"

Response letter to Reviewer #1

General comments

In this study the authors compare the DNDC model simulations of N2O and CO2 emissions against measurements across three different landuse types (arable cropping, grassland, forest). This comparison against different land-use types, as well as the multi-objective Bayesian model calibration method should be of interest to other researchers in this area.

One weakness of this study is that there is no attempt to verify the model calibrations with a independent data. Given that there were three years' data it would have been possible to use two years for the parameterisation and then test the model using the third year's data. (Although this might not work so well for the arable land where the crop types change between years).

We agree with the reviewer in this point . Conducting a split time series calibration and validation test would have been ideal. However, our simulations ran from Jan 2010 to Jan 2016 while measurements for model testing are only available for two years (from Nov 2013 to Dec 2015, see Figure 5, we added this information in section 2.1). We therefore decided not to split up the available 24 months data set, as this would have resulted in a very short period for model testing. In general, it would be very nice to investigate how many data points are needed to test such a complex model set up as we used and how much uncertainty is introduced by reducing the calibration period. For the moment, we cannot answer this question.

Another problem is that the CO2-equivalent values of N2O emissions are incorrectly calculated. It should be noted that the GWP values convert from kg N2O to kg CO2, not kg N2O-N to kg CO2-c.

We acknowledge this important comment. We corrected the GWP values throughout the manuscript.

This MS would benefit from editing by a native English speaker as there a numerous grammatical errors.

We sent the MS to the Nature Research Editing Service. Native English-speaking editors revised it for English language usage, grammar, spelling and punctuation.

Also, results should be consistently presented with their corresponding errors.

We included the communication of the corresponding errors consistently throughout the text.

Specific comments

2.2 Trace gas measurements

- Were any measurements other than CO2 and N2O made?

Yes, there are other measurements like soil moisture, wet deposition, and meteorological data as outlined in chapter 2.1. We decided to combine both chapters to have a stringent structure.

2.3 Modelling approach

- This section contains very little information about how the LandscapeDNDC model works. What processes does it consider, what timestep does it work at, what drivers does it consider? Is it 1, 2, or 3-dimensional? Later in the MS it is mentioned that the model does not consider lateral flows of water and nutrients. This should be mentioned here as many readers might expect that a "Landscape" model would consider horizontal flows.

We agree with the reviewer. We added the following information into this chapter:

"The biogeochemical model MeTr$^x$ simulates the turnover of soil organic matter and plant debris depending on their chemical structures (e.g., lignin and cellulose content, C/N ratio), soil properties (e.g., pH value) and meteorological drivers. Following the 'anaerobic balloon' concept of Li et al. (2000), major metabolites

(e.g., $NO_3$) are distinguished between aerobic and anaerobic counterparts in order to simulate the share of nitrification and denitrification and the related production of GHG emissions. Simulated model outputs are, among others, emissions of $CO_2$ and $N_2O$. The watercycleDNDC model simulates soil water dynamics, i.e., potential evapotranspiration based on Thornthwaite and Mather (1957), transpiration depending on gross primary productivity, the water use efficiency of the modelled plant types and soil water flow based on a cascading bucket model approach (Kiese et al., 2011). The latter determines the advective transport of nutrients into deeper soil layers.

All models refer to a one-dimensional soil column, i.e., assuming homogeneous conditions in lateral directions, and were run with a daily time step resolution."

- pg 5, line 3: If only G1 can be modelled then G2 should not be included in Tables 2 and 3. If G2 is sufficiently different from G1 then it does not make sense to average the results for these two systems
We see the problem with Table 2 and 3. However, we think it makes sense to report the measured fluxes of G2. Instead of deleting this data, we decided to split the rows/columns in Table 2 and 3 for G1 and G2 and explicitly state again, that G2 is not part of the modelling part to prevent confusion.

Table 1
- A column with management practices applied (e.g. manure and fertiliser applications) would also be informative
We like the reviewer's idea and added information about management practices in Table 1.
- Why are the ranges for organic C and N expressed as high-low when all the other ranges are low-high?
We added the missing explanation in the table header: "In case spans are given, they reflect observed ranges for measurements used throughout the set up of the soil profile, given from the top layer setting to the bottom layer of the model."

3.1 Measured N2O fluxes
- pg 6, line 27: Need to clarify what you mean by "no statistical difference over time". Do you mean no differences in the annual cumulative emissions?
We added the following details: "There were no significant differences over time between the three weekly measured transects on arable land (Table 2)". The header of Table 2 explains the underlying statistical test.

- pg 6, line 27: "Highest emissions occur after management events". This is not always true. For example, between May and Sept 2014 there's a measured peak several months after the last management practice
Changed to: "The highest emissions occur mostly after management events."

- pg 6, line 30: CO2e incorrectly calculated. 4.5 kg N2O-N = 7.07 kg N2O (4.5 * 44/28) = 2107 kg CO2e (7.07 * 298) = 575 kg CO2e-C (2107*12/44)
We thank the reviewer for detecting this mistake. We corrected the calculation accordingly throughout the manuscript.

- Grassland N2O: Given that the grazed site and the wetland have different management, soil and vegetation properties it seems strange to combine them both as a single "grassland" type.
We are in line with the reviewer and separated the description of G1 and G2 into the subchapters "Grassland N2O fluxes" and "Wetland N2O fluxes". To be consistent, we did the same in chapter 3.2.

- pg 7, lines 21-22: Conversion of emissions to CO2-Ce is incorrect (see comment for pg 6, line 30)
Changed accordingly.

- pg 7, line 22: How was the emission factor of 5.4% calculated? According to Table 4 the manure N input was 7.57 kg N/ha/a. Assuming no background emissions this gives an emission factor of 0.29/7.57 *100% = 3.8%
Our emission factor value was based on older results, the reviewer is right: the correct emission factor is 3.8%. We changed it accordingly.

- pg 7, line 7: The percentage differences between the forest transects are incorrectly stated. For example, W1 has 3x the emissions of W2, but this is a difference of +200% not +300%. Similarly for the difference between W3 and W2
Changed accordingly.

- pg 7, line 30: The measured negative fluxes are all small compared to the measurement error. Therefore, how do you know that the negative fluxes are real and not just the result of measurement error?
We thank the reviewer for this comment and added it into the manuscript: "However, our measured negative emissions are low compared to the variance between transects (W1-3), i.e., they could also originate from measurement errors."

- pg 7, line 33: Figure A3 shows the WFPS, but it doesn't the correlation between negative emissions and WFPS
We changed the text to: "Negative emissions occur during times with high WFPS (Fig. A3)."

- pg 7, line 34: Conversion of emissions to CO2-Ce is incorrect (see comment for pg 6, line 30)
Changed accordingly.

- pg 8, line 1: 0.08 is almost two orders of magnitude smaller than 5.1
Changed accordingly.

Table 2
- The A3:A3 and G1:G1 squares should be blocked out
Changed accordingly.
- G1 and G2 should be separated into two separate categories (grassland and wetland)
Changed accordingly.

3.2 Measured CO2 fluxes
- I disagree with the decision to use different definitions of 'CO2 emission' depending on land use. I think it would be clearer to use distinct terms such as TER and belowground respiration to avoid the potential for reader confusion
We accept the reviewer suggestion. We explicitly differentiate TER and belowground respiration values now throughout the manuscript, instead of defining them all as CO2 emissions.

- In general measured values should be quoted with uncertainties in this section
Added accordingly.

- pg 8, line 21-22: It is confusing to talk about a weekly measured value and then give the units in day^-1
We removed the word weekly.

- pg 8, line 21-22: What are these results "not significantly different" from?
We added the missing information: "are not significantly different between the transects A1-3 (compare Table 3)."

- pg 9, line 5: What were the measured CO2 fluxes negatively correlated with?
We added the missing information: "with WFPS".

Table 3
- The measurements from G1 and G2 should not be averaged as these systems were sufficiently different that only G1 was able to be modelled. Also, Table 2 explicitly states that G2 was not modelled. This should also be stated in Table 3
Changed accordingly.

Figures 3 and 6
- For the arable and grassland sites the total DNDC results contain values higher than any that occur in the individual seasons. How is this possible?
The "Total" column was showing all simulations of DNDC, while the season columns were showing simulation results only where measurements were available. We changed the "Total" column to only show simulations for days where measurements are available. Accordingly, the revised column does not show any higher values as for the individual seasons.
- It is confusing to include G2 in the grassland results as the model was only run for the G1 system
We removed G2 from Figures 2 and 6.

3.3 Modeled N fluxes
- pg 10, line 17: The uncertainty in NO3- leaching is actually the largest in Table 4
The reviewer is right. We corrected the text at this point.

- pg 11, line 1: Need to be more specific about what you mean when you say emissions were highest in spring. Are you talking about the total, mean, median, variance, upper quartile, or maximum of the emissions?
We added the missing information by stating that we mean the total emissions.

- pg 11, line 16: It is not clear what is the simulated N loss large in comparison to? It could be the grassland N inputs or the losses from the arable system.
We are now more specific at this point. The text reads now: "The simulated N loss is substantially larger than the N input …"
- pg 11, line 19: Do you have any evidence of what is happening to organic N stock in the real system?
We do not have any evidence of the N stocks in the soil so far. We added this information into the manuscript: "The model suggests decreasing soil organic N stocks. So far, we have only initial measurements of soil organic N content. However, we assume that the source of additional N in the form of nitrate in shallow groundwater…"

- pg 11, 19-20: It does not make sense to say that the model is mimicking an additional N source that is not included in the model. The model can only simulate what has been included in the model.
We deleted this statement.

- pg 11, line 26-29: What stocking rate was used for the grazing? Note that for grazed systems the emissions will be spatially as well as temporally peaky. In the grazed system the animal urine patches will create emissions hot spots. With only 5 chambers it is possible that the measurements could miss these hot spots. Meanwhile, the DNDC model will assume that the manure is uniformly spread over the field, producing emissions that are likely to be higher than those from non-urine patches, but lower than those from urine patches.
We thank the reviewer for this point. We added this discussion and the stocking rate (n=70 sheep per hectare) into the manuscript.

- pg 11, line 34: Total output is 1.82, total leaching is 0.04, therefore leaching is 0.04/1.82 *100% = 2% of output.
Changed accordingly.

- pg 12, line 7-9: Measured range 0.18-0.48 kg N2O-N/ha/a does not overlap the measured range of 0.03-0.09 kg N2O-N/ha/a.
The reviewer is right, we changed the text accordingly: "The mean modelled annual emissions $(0.33 \pm 0.15$ kg N ha$^{-1}$ a$^{-1})$ overestimate the observed emissions on all transects."

Table 4
- Should the forest NH3 emission be "<0.01" rather than ">0.01"?
Yes, changed accordingly.

3.4 Modeled C flux
- pg 14, line 12-13: Not sure what the relevance of Figure 7 is here. The statement "mean modeled fluxes are substantially lower than measured ones" contradicts the results in Table 3
We added the missing information with regard to Figure 7: "…mean modelled fluxes are substantially lower than those measured before and after the harvest (Fig. 7)."

- Arable C cycle: There will be some confounding effects in the before/after tillage and before/after harvest emissions in Figure 7. Unless there is a >2 week gap between harvest and tillage the "pre-tillage" results will include some post-harvest effects and the "post-harvest" results will also include some post-tillage effects. Some discussion of how the model handles tillage and harvest events might be informative here.
We agree with the reviewer that e.g., harvest and tillage effects on soil respiration are not distinguishable by sharp points of time. We added the reviewers concern about confounding effects and added a discussion about the model behavior: "However, unless there is a gap of two weeks or more between harvest and tillage, the "pre-tillage" results will include some post-harvest effects, and the "post-harvest" results will also include some post-tillage effects. Our intention to present the data grouped by these events are the discrepancies between modeled and observed $CO_2$ dynamics. There is a sharp drop of modeled $CO_2$ emissions after harvest due to the prompt absence of autotrophic respiration. In reality, there will likely be some ongoing metabolic respiration of plant tissue remaining in the field, which is not represented by the 'assumed' dead plant material in the model. After incorporation of harvest residues (by tilling) modeled $CO_2$ emissions increase again sharply. The sharp increase is due to the incorporation and hence availability of fresh litter (stubble) and a temporary stimulation of decomposition by the model due to the disruption/aeration of the soil structure. Both, overestimation of fresh litter and/or stimulation of decomposition by the model may contribute to the discrepancies between observed and modelled $CO_2$ emissions."

- pg 15, line 6: it is a little odd to describe increasing soil C as an "output" as the C is remaining within the system
Changed to "The rest is related to grazing…"

- pg 15, line 6: Include uncertainties here. In particular it is important to note that the model cannot determine whether the system is net gaining or losing carbon.
We included the uncertainties and the statement of the C balance accordingly.

Table 5
- Should DOC leaching be "<0.01" rather than ">0.01"?
Yes, changed accordingly.

Conclusion
- Table 6: New results shouldn't be presented in the conclusions. This Table should be in the Results and discussion. There also needs to be an explanation of how the model performance was classified as good, medium, or poor.
We see the reviewers point. However, we do not think that Table 6 presents any new results. It is just summarizing the results of chapter 3.3 and 3.4, which is why we would like to keep this table in the

conclusion chapter. We included an explanation how the model performance was classified in the text and in the table header.

- Include uncertainties with results
Changed accordingly.

- It is uncertain whether the grassland was acting as a sink or source of C as the balance 1.35 +/- 4.74 t C/ha/a. Therefore both positive and negative values are within the uncertainty range.
We added this statement in the manuscript. The text reads now: "Whether the extensive grazed grassland is also acting as a sink for C with $1.4 \pm 4.7$ t C ha$^{-1}$ per year remains uncertain …"

Technical corrections
pg 2, line 14-15: Models are not "driven by" uncertainties. Might be better to use "with"?
Changed as proposed.
pg 2, line 15-16: Revise to "During model application further uncertainties arise due to the uncertainties in the applied forcing data"
Changed as proposed.

pg 6, line 24-25: reference to (Fig. 3), (Fig. 4), and (Fig. 5) should be (Fig. 2), (Fig. 3), and (Fig. 4) respectively
Changed accordingly.

pg 6, line 25: Table 4 is referred to before Table 3
We deleted the reference to Table 4 at this point.

pg 7, line 28: "contribute" should be "attribute"
Changed as proposed.

pg 8, line 20: "N" should be "C"
We deleted the reference to Table 5 at this point.

pg 14, line 10: Delete "perfect"
Changed as proposed.

pg 14, line 17: There is no section 3.1.2
Changed to "chapter 2.1".

---

## Author Comment (AC2) · 7 Jun 2017

(comments of the referees are printed in blue, responses of authors are held in black)

We would like to thank the reviewer #2 for his highly constructive comments on the manuscript bg-2017-96 "Constraining a complex biogeochemical model for multi-site greenhouse gas emission simulations by model-data fusion"

**Response letter to Reviewer #2**

The authors used a model-data fusion method to constrain the LandscapeDNDC model with multiple-year greenhouse gas fluxes from arable, forest and grassland sites. I appreciate the authors' effort to parameterize biogeochemistry models using long-term field data. I would like the authors to further clarify some of my concerns:

(1) Is LandscapeDNDC a multi-layer (i.e., vertically-resolved) model? If yes, how many soil layers are included and how do you model the fluxes between layers?

The reviewer's assumption is correct, LandscapeDNDC is vertically-resolved. We included the missing information about the number of layers (arable=85, grassland=40 and forest=45) into the description of the model set up (chapter 2.3.1) and added more details on how the fluxes are modeled: "The watercycleDNDC model simulates soil water dynamics, i.e., potential evapotranspiration based on Thornthwaite and Mather (1957), transpiration depending on gross primary productivity, the water use efficiency of the modelled plant types and soil water flow based on a cascading bucket model approach (Kiese et al., 2011). The latter determines the advective transport of nutrients into deeper soil layers.

All models refer to a one-dimensional soil column, i.e., assuming homogeneous conditions in lateral directions, and were run with a daily time step resolution."

(2) In Tier I and TierII, do you include the same parameters, e.g., all parameters in Table A1? To my understanding, hydrological parameters (e.g., in wcDNDC module) controlling soil water (represented by WFPS) in Teir I will also influence N2O and CO2 in Teir II; however, some key parameters governing the biogeochemistry processes (e.g., in METRX module) do not necessarily affect soil water.

We investigated this potential effect and found no major differences between the WFPS simulation between tierI and tierII. We included this statement into the manuscript and added also some information, which parameters of Table A1 were used in tier I and II.

(3) page 6: please further explain "within best 5% of all simulated RMSEs". Do you mean the best 5% of total number of simulations or the best 5% of unduplicated RMSEs? If it's the former, the number of accepted model runs depends on the total number of model runs. Since multiple objectives (e.g., WFPS in different depths) are considered, do you integrate them (RMSEs) into one single objective? If not, how do you determine the acceptance of a model run?

We added the missing information in the manuscript. The text reads now: "This time, we considered the best 5% of all RMSEs in terms of the respective $N_2O$ and $CO_2$ emissions for each land use (A1-3, G1 and W1-3).".

(4) Table 6: what are the criteria used to classify model performance?

The criteria result from a subjective classification of the model performances compared with each other. We added this information in the table header and where it is referenced in the manuscript.

Other minor comments:

(5) Table 1: Since the model DNDC include inorganic N, what's the inorganic N amount used in the model?

The model considers various inorganic N species ($NH_3$, $NH_4$, $NO_3$, $NO$, $N_2O$, $NO_2$). Respective amounts are dynamically calculated during the simulation and depend strongly on field management such as fertilizer (i.e. inorganic nitrogen) application, which we added in Table 1. Regarding model initialization Tab. 1

presents initial values of total soil nitrogen (inorganic and organic). We changed the term accordingly. However, organic soil nitrogen represents by far the dominating initial N pool (>99%).

How do you initialize the model? Using measured values (e.g., Table 1) or implementing model spin-up?
We added the missing information in the manuscript. The text reads now: "We run simulations for all land uses at a daily time resolution for 6 years, starting on 1$^{st}$ January 2010, using the data from Table 1 as initialization and using a model spin-up time of two years."

(6) please use month names (e.g., January, February,...) to indicate the month in a date (e.g., 01.11.2013)
Changed as proposed.

---

## Author Response (AR2)

(comments of the editor are printed in blue, responses of authors are held in black)

We would like to thank the editor for his highly constructive comments on our manuscript bg-2017-96 "Constraining a complex biogeochemical model for multi-site greenhouse gas emission simulations by model-data fusion"

Comments to the Author:
Dear authors,

many thanks for the thorough revisions, which have addressed the reviewer concerns. I find that the manuscript has been significantly improved. However, I have a couple of minor items that need to be addressed before the manuscript can be accepted for publication.

Best wishes,
Sönke

Comments:

- title: can you think of a more telling / concrete title (e.g. mentioning CO2 and N2O, the land cover types, and/or the geograhic region)? While your study a generally applicable concept of a case-study, the results are specific to the study location.
We changed the title to "Constraining a complex biogeochemical model for CO2 and N2O emission simulations from various land uses by model-data fusion".

- I'm confused with the inclusion of G2: I've search the document several times for G2 to find a reason why it was not modelled. If I've missed this in the text, it may lead better highlighting. However, given that the title of the paper is constraining a model (and not on GHG fluxes in a Hessian landscape), I don't see why these data need to be included at all, if these data are not used for the model fusion, or highlighted in discussion and conclusion as an important area for model development?
We decided to delete G2 from the manuscript.

- I think the conclusion section (or an additional Discussion section) needs to be extended to briefly explain what this study has contributed in terms of novel understanding of GHG fluxes at the study location (at least in terms of potential for new knowledge now that the system works). Currently, the conclusion section is a good summary of the manuscript, but ...
We extended the conclusion chapter with the following outlook: "The herein presented novel GHG study catchment enables a number of future studies. The forest sites could be further used to investigate the influence of leafs on the concentration of N through fall. The presented grassland dataset allows to quantify the nitrate uptake of riparian zones in more detail, e.g., by model coupling analysis, as done by Klatt et al. (2017), to account for potential interactions of land use patterns. Such a model setup would allow upscaling in space, e.g., for the generation of GHG inventories or an analysis of more detailed management scenarios in time. Under the viewpoint of eutrophication and drinking water security, the presented agro-ecosystem plays a pivotal role, as it receives high amounts of reactive nitrogen (N) in form of mineral fertilizer and manure. Our measured data can lead to novel understanding, how to develop and test mitigation measures to reduce N pollution on the landscape scale. The existing model setup can be further used in a forecast mode, e.g. to estimate optimal timing and location of fertilizer application to minimize $N_2O$ emissions and $NO_3^-$ leaching, while at least maintaining yields. Continuous measures of greenhouse gas emissions can be used to evaluate possible mitigation measures."

- p3 L11: Light beech or European/common beech?
Changed to: "young European beech"

- p3 L 18-19: Please check these numbers. Is wet deposition really that low? What about dry deposition?
Note the unit still needs to be kg N / ha / yr.
We came up with these wet deposition values by using measured mean annual rainfall and measured annual mean N concentrations data. Now, by using a more realistic time dependent linear regression to generate daily data, we come up with higher values: 2.70 kg N ha$^{-1}$ a$^{-1}$ and 4.32 kg N ha$^{-1}$ a$^{-1}$ for nitrate and ammonium, respectively. We changed the text at the respective parts of the manuscript and Table 4 (where these values are used) accordingly.
Additionally, we added a statement to the dry deposition: "Dry deposition of N was not measured and can be assumed to add another 30-60% to total atmospheric N deposition (Flechard et al., 2011)."

- Section 2.3.1 - given the tiered model-data-fusion later, briefly explain already.
here whether or not CO2/N2O losses feedback (and if so how) on water cycle and plant physiology
We added the following statement in this section: "It exists an effect of the simulated gaseous losses on the available nutrition's for plant growth, which in turn effects water cycle through changing water uptake and transpiration. However, we neglect this comparably minor interaction effects in our model-data fusion approach, as outlined in section 2.3.2."

- Section 2.3.2 - please introduce the meaning of tier 1 and 2 already at the beginning. Otherwise, this is still hard to follow (ie. tier 1 = water and plants, tier 2, GHG given tier 1).
We added the following statement at the beginning of this section: "Tier I is designed to constrain the investigated parameter space for simulating water cycle and plant growth, while tier II builds up on tier I and aims to fit the parameters for the biogeochemical process, which drive the GHG emissions."

page 7 L 25: define "major effect"
We added: "…, i.e. simulated soil moisture did not change substantially with changing biogeochemical model parameters."

page 10 L13 (and possibly similar instances): four numbers are given, but associated with three treatments, which does not work out. Please clarify text (here, and possibly at similar occasions).
Corrected accordingly.

in general: avoid the use of "quite" low etc. thorughout the manuscript. Please use exact language.
Changed as proposed.

Please revise figure axis label to be larger and thus easier to decipher.
Changed as proposed.